# Adv-BMT: Bidirectional Motion Transformer for Safety-Critical Traffic Scenario Generation

**Yuxin Liu**\* **Zhenghao Peng**\* **Xuanhao Cui** **Bolei Zhou**
University of California, Los Angeles

## Abstract

Scenario-based testing is essential for validating the performance of autonomous driving (AD) systems. However, such testing is limited by the scarcity of long-tailed, safety-critical scenarios in existing datasets collected in the real world. To tackle the data issue, we propose the Adv-BMT framework, which augments real-world scenarios with diverse and realistic adversarial traffic interactions. The core component of Adv-BMT is a bidirectional motion transformer (BMT) model to perform inverse traffic motion predictions, which takes agent information in the last time step of the scenario as input, and reconstruct the traffic in the inverse of chronological order until the initial time step. The Adv-BMT framework is a two-staged pipeline: it first conducts adversarial initializations and then inverse motion predictions. Different from previous work, we do not need any collision data for pretraining, and are able to generate realistic and diverse collision interactions. Our experimental results validate the quality of generated collision scenarios by Adv-BMT: training in our augmented dataset would reduce episode collision rates by 20%. Demo and code are available at `https://metadriverse.github.io/adv-bmt/`.

## 1 Introduction

In recent years, autonomous driving (AD) agents have achieved unprecedented performance in simulations [3, 9, 23]. However, handling corner traffic situations, especially collision scenarios, remains a major challenge. A major cause is that safety-critical scenarios are missing from real-world driving datasets due to high costs and risks of data collections. Without enough collision training data, it is hard for the autonomous driving (AD) planners and prediction models to learn safe driving in challenging and risky scenarios. This motivates the need for simulating different real-world accidents. To generate realistic collision trajectories, previous works [15, 27, 18] leveraged learned real-world traffic priors, and optimized predictions on collision-encouraging objectives. However, our evaluations of these baselines reveal that the generated behaviors are insufficiently diverse and yield a low collision generation rate.

We tackle this challenge by rethinking the motion prediction architecture itself. We introduce the Bidirectional Motion Transformer (BMT), a new model that learns to predict both future and history trajectories for all agents, conditioned on their current states. Similar to recent autoregressive traffic forecasting models [24, 16, 30], BMT tokenizes continuous trajectories into discrete control actions. Distinct from prior work, BMT employs a temporally reversible tokenization scheme that enables unified forward (future) and reverse (history) motion prediction within the same framework.

We utilize BMT model into our Adv-BMT framework for realistic and diverse collision traffic generations from real-world driving data. While existing works follow a standard paradigm, which selects a convenient neighbor agent and modifies the behavior to attack the ego agent, Adv-BMT inserts new

---

\*Equal contribution

39th Conference on Neural Information Processing Systems (NeurIPS 2025).

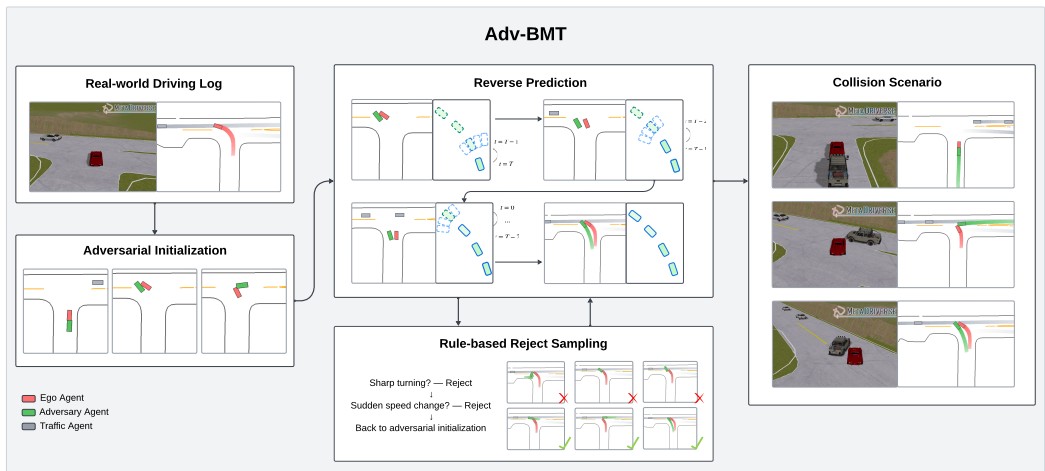

Figure 1: **Overview of Adv-BMT.** The framework mainly consists of three steps: (1) it adds an adversarial agent (ADV) with a sampled collision state into the current scenario; (2) it predicts reversely for the adversarial trajectory; (3) it performs rule-based checks and rejects physically implausible ones.

agents (ADV) with diverse collision interactions, maintaining realistic interaction with other traffic agents. In short, Adv-BMT is a three-staged pipeline: first, it initializes diverse collision frames between a new adversary agent (ADV) and ego vehicle; then, it reconstructs the adversarial trajectories via BMT's reverse predictions; finally, it conducts rule-based checks and rejects trajectories with physically implausible collision initializations. An overview of our framework is illustrated in Fig. 1. It is worth noting that that there is no collision data included in our model training.

Adv-BMT is designed with multiple generation modes to support varying levels of agent interactions and realism. By default, traffic agents follow their recorded trajectories to preserve consistency with the real-world driving log, while the adversarial agent is generated to interact within this fixed context. This enables targeted scenario editing while maintaining overall scene plausibility. To address limitations in interactivity, Adv-BMT also supports a closed-loop reverse prediction mode, in which all agents are jointly predicted to generate a fully reactive traffic scenario, without teacher-forcing any agent behaviors during predictions. Additionally, an optional forward refinement step allows traffic agents to respond to the newly introduced adversary, enabling a more interactive and dynamically consistent outcome. Together, these modes allow Adv-BMT to balance realism, controllability, and diversities for different use cases.

We summarize our contributions as follows: (1) We introduce the bidirectional motion transformer with temporally reversible motion tokenizations; (2) We develop Adv-BMT for realistic and diverse safety-critical traffic simulations; (3) We leverage Adv-BMT in a closed-loop setting to dynamically create challenging environments for reinforcement learning agents.

## 2 Related Work

**Driving Motion Prediction.** The task of motion prediction focuses on forecasting the future trajectories of traffic participants conditioned on their initial map context and agent states. Recent work uses transformer models to autoregressive sequence modeling. A set of work utilizes discretized motion tokenizations to perform next token predictions: Trajeglish [12] discretizes motion representations using a k-disk-based tokenization scheme to represent position and angle differences for relative movements. MTR++ [17] directly represents motion on continuous space in Gaussian mixture distributions. MotionLM [16] models trajectory deltas. SMART builds a k-disk–clustered vocabulary of motion tokens, containing coordinates, heading, and shape. BehaviorGPT [30] performs next-patch predictions with future motion chunks, instead of single-step predictions. The BMT model constructs two sets of motion tokens based on inter-frame accelerations, enabling both

forward prediction of future motions and inverse reconstruction of past motions. Another set of works [6, 29, 28, 10, 26, 13, 25, 22] leverage diffusion models for motion predictions.

**Safety-Critical Traffic Scenario Generation.** To generate collision traffic, a classic line of previous works use a two-staged method: first uses a traffic prior to generate realistic agent trajectories, then use collision-sensitive objectives for trajectory optimization or candidate selections. STRIVE [15] models traffic by a variational autoencoders but requires computationally expensive per-scenario optimizations. CAT [27] leverages transformer motion decoder [5] and selects the collision trajectories and simulates in MetaDrive [9] for closed-loop adversarial environments. SEAL [18] uses a skill-based adversarial policy with collision-related objectives. SafeSim [1] proposes a diffusion model with a test-time collision-sensitive guidance loss to control the collision type and adversarial agent selections. Another set of work such as [14] use a reinforcement learning (RL) based approach, which parametrizes trajectories and goal constraints to generate safety-critical interactions. AdvSim [21] directly perturbs actor trajectories using a kinematic model and optimizes via a black-box adversarial loss. Another line of recent work such as [2] uses a conditional normalizing flow to model the distribution of real-world safety-critical trajectories. CrashAgent [7] and LCTGen [19] leverage free-form texts as inputs and extract embeddings to parameterize scene initializations and agent driving directions. Different from previous approaches, Adv-BMT generates collision scenarios in three steps: first samples a collision state, then conduct reverse predictions, finally forcast traffic agents accordingly.

## 3 Method

Classical motion prediction models forecast future trajectories based on the current states of traffic agents. Building on this foundation, we propose the reverse motion prediction problem. To address both tasks within a unified framework, we introduce the Bidirectional Motion Transformer (BMT) model, which is able to perform both tasks. Finally, we present the Adv-BMT framework and describe how it leverages BMT model as the core to generate realistic and diverse safety-critical interactions.

### 3.1 Bidirectional Motion Prediction

We first introduce the bidirectional motion prediction task, shown in Fig. 2. Consider a traffic scenario with at most $N$ agents and a prediction horizon of $T$ steps. The trajectory of agent $i$ is represented as $\boldsymbol{\tau}^i = \{\tau_0^i, \tau_1^i, \ldots, \tau_T^i\}$, where each state $\tau_t^i \in \mathbb{R}^d$ encodes its position, velocity, and heading at time $t$. We introduce a prediction direction indicator $D \in \{\text{Forward}, \text{Reverse}\}$, to specify whether the model predicts future or past motion over the horizon.

For each predicted traffic agent, we construct a sequence of motion tokens $\mathbf{Z}^i = \{z_1^i, \ldots, z_T^i\}$, by applying the motion tokenization function $\phi(\cdot)$ between consecutive states. In the forward setting, the tokens are derived in chronological order, from $\tau_0^i$ to $\tau_T^i$. In the reverse setting, the temporal order of the trajectory is inverted, and the tokens are generated by applying $\phi(\cdot)$ backward from the current state toward the initial state. This formulation yields a bidirectional token sequence that enables BMT to model both forward and reverse motion dynamics in a unified token space. The tokenization function $\phi(\cdot)$ is further detailed in Section 3.2.

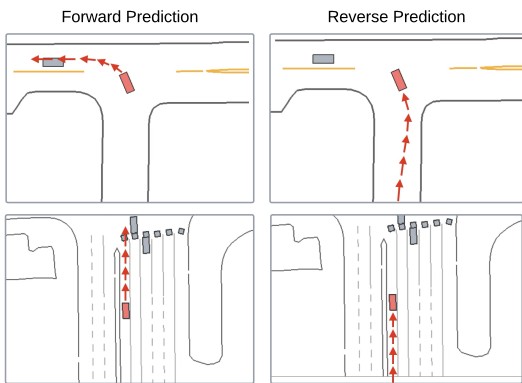

Figure 2: **Bidirectional predictions on the ego agent (red).** BMT supports predictions for future motions (left) and historical motions (right) jointly for all prediction agents.

### 3.2 Bidirectional Motion Transformer (BMT)

**Token Space.** BMT's bidirectional motion tokens are derived from a simplified bicycle dynamics model. Both forward and reverse tokens are defined over the same shared token space—a set

of discrete bins of acceleration and yaw rate pairs. We uniformly quantize the control space of accelerations $a \in [-a_{\max}, a_{\max}]$ and yaw rates $\delta \in [-\delta_{\max}, \delta_{\max}]$ into $K$ bins each, yielding a total of $K^2$ discrete motion tokens, where $a_{\max} = 10 m/s$, $\delta_{\max} = \frac{\pi}{2}$, and $K = 33$.

**Motion Token Reconstruction.** A key step in BMT is to reconstruct continuous trajectories from discrete motion tokens, bridging between physical motion and the token vocabulary. Each token encodes a low-level control command defined by an acceleration $a$ and a yaw rate $\omega$. Starting from the current agent state $\tau = (x, y, \theta, v)$, the token is applied over a time interval $\Delta t$ to update the agent's position, heading, and velocity. BMT adapts midpoint integration to propagate the state, which provides a stable and realistic approximation of vehicle dynamics. Intuitively, the speed and heading are updated using their midpoint values, ensuring that the reconstructed trajectory remains smooth and physically consistent. This reconstruction procedure allows BMT to decode token sequences into continuous motion trajectories in both forward and reverse prediction settings.

**Model Architecture.** The BMT architecture overview is shown in Fig. 3. BMT has a scene encoding component used to obtain embeddings for scenario contexts with separate embeddings for map polylines, traffic lights, and agent initial states. Then, we use Fourier-encoded edge features [20] to represent the spatial and directional information between these encoded entities, which are then passed to the transformer encoder with self-attention layers for the relational embeddings.

The prediction decoder predicts subsequent motion tokens in an autoregressive manner, with only initial agent information for the first frame, along with the scene embedding obtained from the Scene Encoder. The motion decoder incorporates self-attention over the initial agent token embeddings, and three relation computations separately: agent-to-agent (a2a), agent-to-time (a2t), and agent-to-scene (a2s), with each relation embedding then passed to its cross-attention layers. The output agent embeddings are concatenated and repeated a number of times. The output agent motion embeddings are mapped to the vocabulary of discretized motion tokens through MLPs. More details can be found in the Appendix.

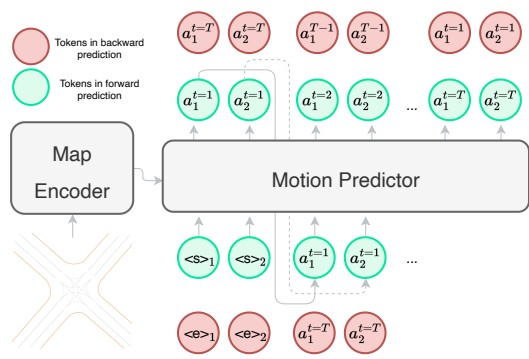

Figure 3: **BMT architecture.** BMT consists of a scene encoder and a GPT-style motion decoder. It employs two sets of motion tokens for forward and reverse predictions to generate the next-step token for each agent. All predictions are conditioned only on the map information and the one-step current state of all predicted agents.

**Training.** BMT is trained to learn a policy that reproduces the distribution of real-world driving behaviors. At each step, the model predicts a discrete motion token for every agent based on its past tokens, current state, and the scene context. To align these predictions with the ground-truth behaviors in the dataset, we minimize a cross-entropy loss between the predicted and observed token distributions:

$$\mathcal{L}_{\text{train}} = - \mathbb{E}_{\mathcal{D}} \left[ \sum_{t=1}^{T} \sum_{i=1}^{N} \log \pi_\theta \left( z_t^i \mid \mathbf{z}_{1:t-1}^i, \mathcal{M} \right) \right], \quad (1)$$

where $\pi_\theta$ is the token prediction policy parameterized by $\theta$. Intuitively, this objective encourages the model to assign high probability to motion tokens that correspond to real trajectories, thereby capturing the joint distribution of multi-agent actions in traffic.

During inference, BMT generates motion tokens autoregressively, sampling one token at a time conditioned on its previous predictions. We apply nucleus (top-$p$) sampling to promote behavioral diversity while remaining faithful to the learned distribution. To mitigate exposure bias, the model rolls out on its own sampled tokens rather than the ground-truth sequence, ensuring consistency between training and inference.

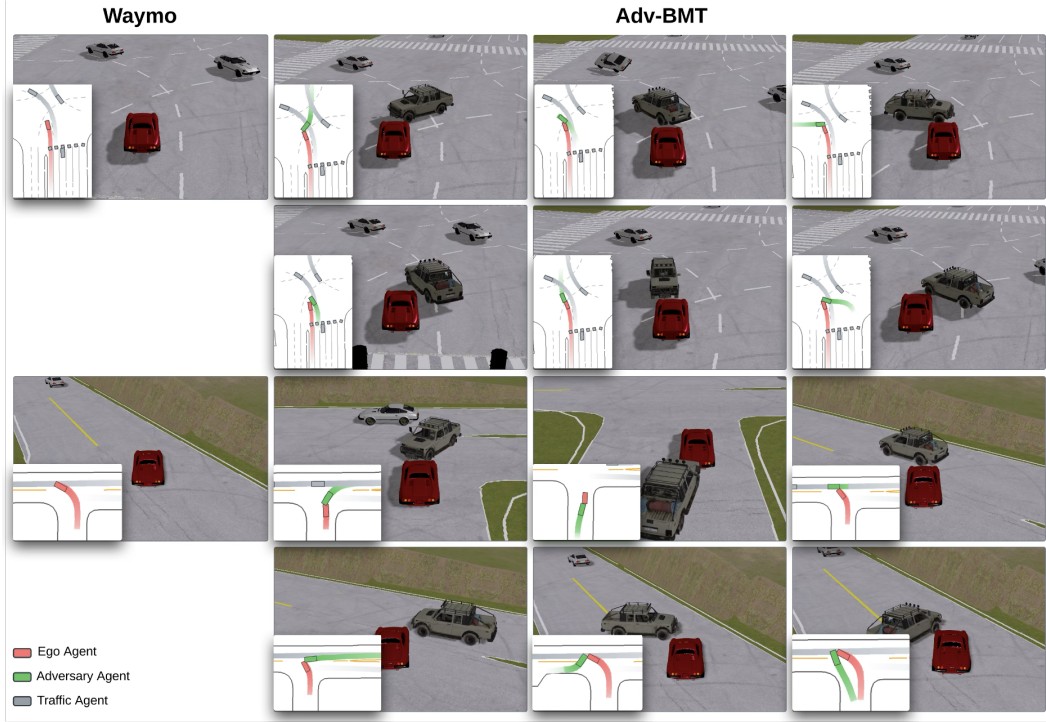

Figure 4: Diverse collision headings generated by Adv-BMT from the same driving log input, visualized from a third-person perspective.

### 3.3 Adv-BMT for Safety-critical Interaction

The BMT model serves as the core that captures the distribution of realistic multi-agent motions in Adv-BMT framework. Building upon this foundation, we develop Adv-BMT, an adversarial scenario generation framework designed to generate realistic and safety-critical traffic interactions. The overview of Adv-BMT framework is illustrated in Fig.1. From the input scenario, Adv-BMT first samples diverse collision states between ego agent and a new adversarial agent (ADV), then reconstructs complete trajectories via reverse motion predictions. A rule-based reject sampling mechanism is used for selecting collision initializations.

**Diverse Adversarial Initialization.** While existing works select a convenient neighbor agent and utilize its initial information, Adv-BMT inserts new agents (ADV) with diverse collision initializations for different opponent interactions. We visualize an example result in 5. Formally, we define the collision state as to include the position, time, velocity, and heading at the collision step. The collision time can be varied, sampled from the first second to the last time step of the ego trajectory length. Similarly, ADV's collision headings are randomly sampled. ADV's collision position can be calculated from collision heading and the ego vehicle's collision position. Last but not least, collision velocity is calculated from sampling a offset from ego vehicle's speed at collision step.

**Multi-agent Adversarial Interaction.** A key advantage of Adv-BMT lies in its ability to gen-

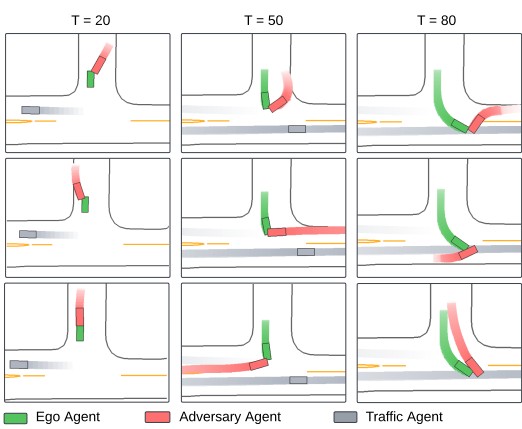

Figure 5: Diverse collision timings generated by Adv-BMT from the same driving log input, visualized in a bird's-eye view.

erate realistic and flexible multi-agent interactions surrounding the adversarial agent (ADV). Starting from sampled collision initializations, the ADV is added into the scene together with existing traffic participants. BMT then performs reverse-time prediction to reconstruct plausible interaction histories that lead to the designated collision state. To balance realism, controllability, and scene reactivity, Adv-BMT provides several generation modes with different levels of multi-agent coupling:

1. **Adv-BMT with replayed traffic agents:** In this mode, traffic agents are teacher-forced to follow their original recorded trajectories, while only the adversarial agent's history is predicted in reverse time. This preserves the original traffic flow and ensures high scene fidelity. Teacher-forcing prevents unintended deviations (e.g., traffic agents switching lanes or diverging from the original trajectory) that may occur if the model predicts all agents in the closed-loop manner. As a result, BMT reconstructs the adversarial agent's trajectory to fit into the original scenario.

2. **Adv-BMT with closed-loop reverse prediction:** This mode removes teacher-forcing and jointly predicts the trajectories of all agents, including both traffic participants and the adversarial agent, in a single reverse prediction pass. By allowing all agents to evolve backward in time simultaneously, the scene develops coherently as a fully interactive scenario. This setting enables rich multi-agent dynamics and can produce fully synthesized safety-critical scenarios.

3. **Adv-BMT with forward refinement:** This hybrid mode first applies teacher-forced reverse prediction to generate the adversarial agent, then runs an additional forward prediction pass to enable traffic agents to react to its presence. This introduces interaction and reactivity without discarding the structure of the original recorded scene, offering a middle ground between strict replay and fully closed-loop generation.

These complementary modes provide a controllable trade-off between scene realism, interactivity, and controllability, enabling users to select the most suitable configuration for different evaluation objectives.

**Rule-based Rejection Sampling.** We design Adv-BMT to have diverse initializations, which do not guarantee the realism of the collision outcomes. To address this issue, we implement a rule-based filtering rejection mechanism for ADV candidates. Specifically, we first measure the driving distance and average speed of ADV candidates; if it moves too short and mostly wanders at the designated position waiting for the ego vehicle, then it is considered invalid. Meanwhile, we check the max curvature (the rate of change for heading): given a candidate ADV trajectory, we compute the curvature constraint using: $\kappa_t = \Delta\theta_t/\Delta s_t$, where $\Delta\theta_t$ is the absolute heading change between time steps, and $\Delta s_t$ is the displacement between consecutive positions. Adv-BMT rejects a prediction if $\max_t(\kappa_t) > \kappa_{\text{threshold}}$, where $\kappa_{\text{threshold}} = 0.8$ is a predefined curvature limit that we found useful. We enforce a curvature constraint to ensure that the trajectory remains within the predefined threshold, rejecting ADV candidates that exhibit unrealistically sharp turning behavior. With our straightforward rule-based rejection sampling mechanism, we are able to maintain realistic collision events between the ADV and the ego vehicle.

## 4 Experiments

We first assess the BMT model on its ability to generate realistic and diverse traffic behaviors in Section 4.1. We then evaluate quality of Adv-BMT scenarios compared to three baseline methods in Section 4.2. Furthermore, in Section 4.3 we evaluate the utility for the downstream learning task, and assess whether training a reinforcement learning (RL) planner in Adv-BMT scenarios lead to improved performance and robustness compared to log-replay traffic flows.

**Dataset.** All experiments use driving data from the Waymo Open Motion Dataset (WOMD) [4] with formats managed by ScenarioNet [8]. WOMD contains 10Hz scenarios, each with 1 second of history and 8 seconds of future trajectories. Each scenario includes up to 128 traffic agents including vehicles, cyclists, pedestrians along with high-definition maps. To reduce computational cost, we downsample each scenario to 2Hz, yielding 19 prediction steps. We randomly select 500 scenes for both open-loop evaluation and RL training. We use 6 prediction modes for each scene during open-loop evaluations in Section 4.1 and Section 4.2.

Table 1: We evaluate BMT under three types of prediction tasks in a closed-loop setting for all traffic agents, without replaying any agents. In the bidirectional mode, forward prediction is performed based on the reverse prediction results. The number of prediction modes is set to six.

(a) Realism of BMT predictions.

| Method | $\text{SFDE}_{avg}$ | $\text{SFDE}_{min}$ | $\text{SADE}_{avg}$ | $\text{SADE}_{min}$ | $\text{VehColl}_{min}$ | $\text{VehColl}_{avg}$ | $\text{JSD}_{velocity}$ | $\text{JSD}_{TTC}$ |
|---|---|---|---|---|---|---|---|---|
| Reverse | 3.92 | 2.70 | 5.53 | 5.11 | 0.03 | **0.05** | **0.17** | **0.02** |
| Forward | 3.52 | 2.35 | 2.39 | 1.98 | 0.03 | **0.05** | 0.23 | 0.30 |
| Reverse + Forward | **3.29** | **2.01** | **2.38** | **1.74** | 0.03 | 0.06 | 0.22 | 0.35 |

(b) Diversity of BMT predictions.

| Method | SDD | FDD | ADD |
|---|---|---|---|
| Reverse | 8.35 | - | 3.13 |
| Forward | - | 10.78 | 4.40 |
| Reverse + Forward | **8.49** | **12.78** | **7.00** |

**Metrics.** To evaluate the realism of BMT predictions, we adopt standard open-loop prediction accuracy metrics: Scenario Final Displacement Error (SFDE) and Scenario Average Displacement Error (SADE). For each, we report results on both the best prediction mode and the average over all six prediction modes. We also measure the average number of agent collisions to capture interaction realism. To assess the diversity of generated trajectories, we report Final Displacement Diversity (FDD), Starting Displacement Diversity (SDD), and Average Displacement Diversity (ADD), which quantify the spread of predicted positions across all prediction modes. We also compute the Jensen–Shannon Divergence (JSD) values over velocity, acceleration, and time-to-collision (TTC) distributions between generated and ground-truth trajectories. JSD measures the similarity between two probability distributions, with lower values indicating closer alignment between generated and real-world behaviors.

## 4.1 Evaluation of BMT Predictions

Results from Table 1a indicates that BMT performs realistic scenario generations in forward, backward, and bidirectional predictions. Reverse predictions perform slightly worse than forward predictions in realism metrics. Compared to single-pass prediction, bidirectional prediction exhibit higher prediction accuracy. Despite these differences, the overall collision rate remains reasonable and comparable for all prediction tasks, which indicates that BMT effectively generate realistic traffic interactions. The results in Table 1b show that bidirectional prediction enhances the diversity of generated agent behaviors across all displacement diversity metrics. Forward prediction exhibits higher FDD and ADD scores than reverse prediction. This indicates that predicting future motions encourages more explorations for variant directions.

## 4.2 Evaluation of Adv-BMT Scenarios

In Table 2, we additionally use the adversarial attack success rate (i.e., the collision rate between adversary and the ego), ADV-Traffic Collision Rate (i.e.,collision rates between adversary agents and traffic agents), and the average Agent Collision (i.e., average traffic agent collision rate) to indicate interaction realism. Two settings of Adv-BMT outperform baselines in both realism and diversity. This validates Adv-BMT's design of collision initialization + reverse predictions, which couldn't be achieved by other methods. While Adv-BMT generates highly adversarial behaviors, it also preserves diversity of traffic interactions compared to baselines. FDD and SDD metrics suggest that baselines generate nearly the same adversarial trajectories on the given scene. The JSD metrics suggest Adv-BMT outperforms in realism metrics compared to baselines. BMT model is able to achieve realistic motion predictions indicted by Waymo Open Sim Agent Challenge metrics [11], which we add in Table 6 in our appendix. Furthermore, the results for Adv-BMT with filtering indicate that our rule-based filter does not harm diversity and at the same time enhances the realism metrics. Table 3 demonstrates the generation speed comparisons among all methods. Our evaluations validate Adv-BMT as an efficient framework for realistic, diverse, and safety-critical generation.

Table 2: **Diversity and realism of generated adversarial behaviors.** We compare Adv-BMT scenarios with three baseline methods for safety-critical interactions, namely STRIVE [15], CAT [27], and SEAL [18]. To be consistent with baseline methods in this evaluation only, we force Adv-BMT to choose nearest neighboring agent and modify its behavior as the predicted adversary.

| Method | FDD | ADD | $JSD_{Vel}$ | $JSD_{Acc}$ | $JSD_{TTC}$ | Attack Succ. | Agent $Coll_{min}$ | ADV-Traffic $Coll_{min}$ |
|---|---|---|---|---|---|---|---|---|
| CAT | 0.00 | 0.00 | 0.13 | 0.16 | 0.12 | 0.48 | 0.10 | 0.3 |
| SEAL | 0.00 | 0.00 | 0.18 | 0.25 | 0.17 | 0.55 | 0.12 | 0.32 |
| STRIVE | 0.01 | 0.00 | 0.11 | 0.22 | 0.12 | 0.38 | **0.09** | 0.32 |
| BMT_TF | 1.22 | 0.49 | **0.10** | **0.17** | 0.13 | **1.00** | 0.13 | 0.22 |
| BMT_All | **1.51** | **0.63** | **0.10** | **0.17** | **0.08** | **1.00** | 0.13 | **0.19** |
| BMT_TF + Filter | **2.32** | **1.00** | **0.08** | **0.12** | 0.09 | **1.00** | 0.13 | 0.12 |
| BMT_All + Filter | 1.97 | 0.83 | 0.13 | 0.14 | **0.08** | **1.00** | **0.12** | **0.09** |

Table 3: **Generation speeds across methods.** Among the four methods, CAT achieves the fastest generation speed, followed by Adv-BMT with a slightly lower speed. SEAL and STRIVE are comparatively slower.

| Method | CAT | SEAL | STRIVE | Adv-BMT |
|---|---|---|---|---|
| 500-avg (seconds) | **0.80** | 2.36 | 9.53 | 1.02 |

**Visualization.** We simulate Adv-BMT scenarios in MetaDrive [9], rendering in both bird's-eye view and third-person perspective. In Fig. 5, Fig. 4, and Fig. 9, we show diverse Adv-BMT adversarial behaviors in the collision directions on the same ego agent in several real-world traffic flows from WOMD. Generated adversarial agents follow traffic rules and maintain realistic driving patterns. Adv-BMT supports diverse adversarial agent types including vehicles, pedestrians, and cyclists, as shown in Fig. 6. Adv-BMT makes full use of each driving log, which makes Adv-BMT suitable for AD testing and adversarial training. A visual comparison between Adv-BMT and baseline methods is shown in Fig. 7. When a baseline fails to select an existing traffic agent that is convenient for a safety-critical attack, Adv-BMT is able to imagine a new agent at an appropriate position to perform adversarial attacks on the ego vehicle.

## 4.3 Adversarial Learning

To validate the value of Adv-BMT scenarios in downstream autonomous driving (AD) tasks, we train a reinforcement learning (RL) agent within augmented scenarios containing collision interactions generated by Adv-BMT and baseline methods. To determine the quality of the augmented training scenarios, we measure both the driving performance and the safety performance of the learned AD agent compared to learning on the original training set. In our experiment, we conduct two sets of training: (1) open-loop RL, where the agent is trained on a fixed training set with adversarial scenarios generated based on ground-truth ego trajectories; and (2) closed-loop RL, where an adaptive adversarial agent attacks the current ego agent based on its recent rollout trajectory records. The adaptive adversarial agent's motion is generated by Adv-BMT or a baseline method. Results are shown in Table 4 and Table 5.

**Setting.** The training set contains 500 real-world scenarios randomly selected from the WOMD training set. We train a Twin Delayed DDPG (TD3) agent for 1 million steps using 8 random seeds to ensure robustness in MetaDrive [9] (hyperparameters listed in the appendix). We measure the average reward, average step cost, average route completion rate (Compl.), and average episode cost (cost sum) for driving performance measurement. To evaluate the impact of adversarial training using Adv-BMT-generated scenarios, we assess policy performance across two distinct validation environments: (1) 100 Waymo validation environments, which consist of unmodified real-world driving scenarios from WOMD validation set, and (2) 100 Adv-BMT environments, which is the augmented collision scenarios from the 100 validation scenes.

**Analysis.** Table 4 reports open-loop evaluations comparing RL policies trained on Adv-BMT–generated scenarios against policies trained on baseline scenario sets. On the WOMD original validation environments, Adv-BMT–trained agents outperform all baselines across metrics: episode cost decreases by 10% and collision rate by 8%, while reward and route-completion remain com-

Table 4: **Open-loop RL agent evaluation.** Each WOMD scenario is augmented with one collision scenario, so that the ratio between real-world and safety-critical scenes is 1:1. For each method, we augment the original dataset into a new training set.

(a) Evaluation in the Waymo validation environments

| Training Scenarios | Reward ↑ | Cost ↓ | Completion ↑ | Collision ↓ | Cost Sum ↓ |
|---|---|---|---|---|---|
| Waymo [4] | 32.03 $\pm$ 4.27 | 0.39 $\pm$ 0.07 | 0.72 $\pm$ 0.05 | 0.14 $\pm$ 0.02 | 1.41 $\pm$ 0.35 |
| CAT [27] | 30.37 $\pm$ 3.89 | 0.39 $\pm$ 0.05 | 0.71 $\pm$ 0.05 | 0.14 $\pm$ 0.02 | 1.73 $\pm$ 0.39 |
| STRIVE [15] | 31.30 $\pm$ 3.59 | 0.40 $\pm$ 0.04 | 0.73 $\pm$ 0.05 | 0.13 $\pm$ 0.03 | 1.51 $\pm$ 0.40 |
| SEAL [18] | 29.94 $\pm$ 5.14 | 0.39 $\pm$ 0.05 | 0.71 $\pm$ 0.04 | 0.12 $\pm$ 0.02 | 1.63 $\pm$ 0.44 |
| Adv-BMT | 31.47 $\pm$ 3.21 | 0.38 $\pm$ 0.03 | 0.73 $\pm$ 0.04 | **0.11** $\pm$ **0.02** | **1.35** $\pm$ **0.40** |
| Adv-BMT (Refined) | **33.22** $\pm$ **1.83** | **0.36** $\pm$ **0.03** | **0.74** $\pm$ **0.03** | 0.12 $\pm$ 0.02 | 1.39 $\pm$ 0.22 |

(b) Evaluation in the Adv-BMT validation environments

| Training Scenario | Reward ↑ | Cost ↓ | Completion ↑ | Collision ↓ | Cost Sum ↓ |
|---|---|---|---|---|---|
| Waymo [4] | 37.01 $\pm$ 6.16 | 0.64 $\pm$ 0.09 | 0.60 $\pm$ 0.07 | 0.30 $\pm$ 0.02 | 2.96 $\pm$ 0.63 |
| CAT [27] | 36.77 $\pm$ 4.95 | 0.62 $\pm$ 0.05 | 0.62 $\pm$ 0.05 | 0.29 $\pm$ 0.02 | 3.09 $\pm$ 0.56 |
| STRIVE [15] | 37.72 $\pm$ 5.38 | 0.63 $\pm$ 0.06 | 0.63 $\pm$ 0.06 | 0.29 $\pm$ 0.04 | 2.92 $\pm$ 0.68 |
| SEAL [18] | 35.74 $\pm$ 6.36 | 0.67 $\pm$ 0.08 | 0.60 $\pm$ 0.06 | 0.31 $\pm$ 0.01 | 2.97 $\pm$ 0.34 |
| Adv-BMT | 37.33 $\pm$ 3.57 | 0.62 $\pm$ 0.03 | 0.63 $\pm$ 0.04 | **0.25** $\pm$ **0.05** | **2.41** $\pm$ **0.43** |
| Adv-BMT (Refined) | **39.55** $\pm$ **2.94** | **0.59** $\pm$ **0.04** | **0.65** $\pm$ **0.02** | 0.27 $\pm$ 0.04 | 2.74 $\pm$ 0.54 |

Table 5: **Closed-loop RL agent evaluation.** We use the same augmented training set as in the open-loop experiment. For the adaptive adversarial learning experiment, we implement an adaptive generator for Adv-BMT and CAT [27]. We discard the other baseline methods due to their low generation speeds.

(a) Waymo Validation Environments

| Generator | Reward ↑ | Cost ↓ | Completion ↑ | Collision ↓ | Cost Sum ↓ |
|---|---|---|---|---|---|
| CAT | 32.15 $\pm$ 2.89 | **0.38** $\pm$ **0.03** | 0.74 $\pm$ 0.04 | 0.10 $\pm$ 0.00 | 2.02 $\pm$ 0.24 |
| **Adv-BMT** | **33.13** $\pm$ **4.11** | 0.39 $\pm$ 0.03 | 0.74 $\pm$ 0.03 | **0.09** $\pm$ **0.00** | **1.25** $\pm$ **0.52** |

(b) Adv-BMT Environments

| Generator | Reward ↑ | Cost ↓ | Completion ↑ | Collision ↓ | Cost Sum ↓ |
|---|---|---|---|---|---|
| CAT | 39.47 $\pm$ 3.88 | 0.62 $\pm$ 0.05 | 0.63 $\pm$ 0.04 | 0.22 $\pm$ 0.02 | 2.51 $\pm$ 0.45 |
| **Adv-BMT** | **40.40** $\pm$ **6.39** | **0.57** $\pm$ **0.04** | 0.63 $\pm$ 0.05 | 0.22 $\pm$ 0.04 | **2.48** $\pm$ **0.97** |

parable to the best baseline. Adding the optional forward refinement (Sec. 3.3) yields additional gains—reward increases by 6%, episode cost decreases by 5%, and route completion improves slightly. On safety-critical test environments, Adv-BMT further reduces episode cost by 17% and collision rate by 14% relative to the best baseline. With forward refinement, we observe further improvements in driving capability, reflected in higher reward and completion rates. These results indicate that training on Adv-BMT scenarios exposes agents to a broader distribution of high-risk interactions, enabling more robust, safety-aware policies. The extra gains from forward refinement underscore the importance of modeling reactive traffic around emerging incidents, which better prepares agents for cascading hazards. Motivated by these findings, we next evaluate all methods in a closed-loop RL setting.

Table 5 reports closed-loop RL results under reactive adversarial traffic generators. Relative to the corresponding open-loop policy, the closed-loop policy improves across all metrics in both real-world and safety-critical collision environments, indicating a substantially safer and more robust AD policy. Compared with the strongest closed-loop baseline (CAT), our method achieves higher reward, lower collision rate, and a 38% reduction in episode cost. These gains suggest that training with Adv-BMT scenarios—featuring diverse adversarial behaviors and realistic, traffic-consistent reactions—better train the agent to anticipate and mitigate high-risk interactions. Learning in closed-loop further improves performance; by training the agent to respond online to reactive opponents, it becomes better at anticipating dangerous situations, and staying safe in unfamiliar or rapidly changing traffic conditions. The results also highlight the importance of the Adv-BMT design, which enables the generation of realistic and flexible multi-agent interactions with the adversarial agent.

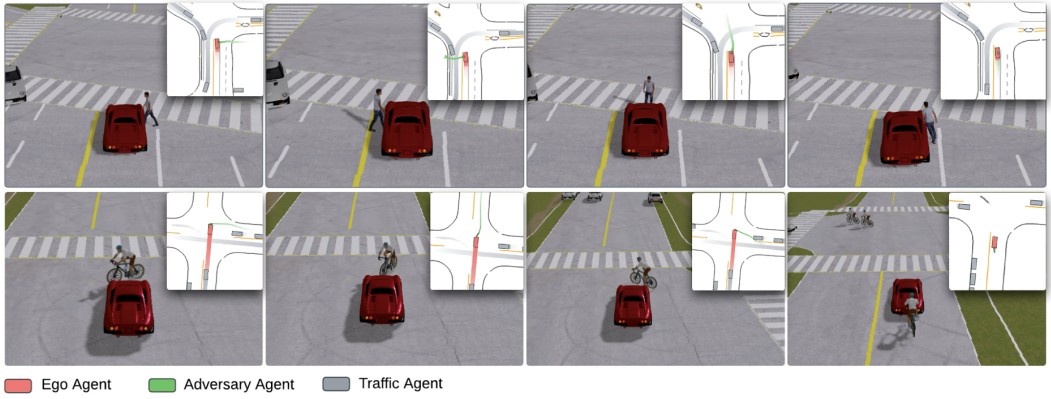

Ego Agent    Adversary Agent    Traffic Agent

Figure 6: Adv-BMT can also generate scenarios of pedestrian and cyclist adversarial agents.

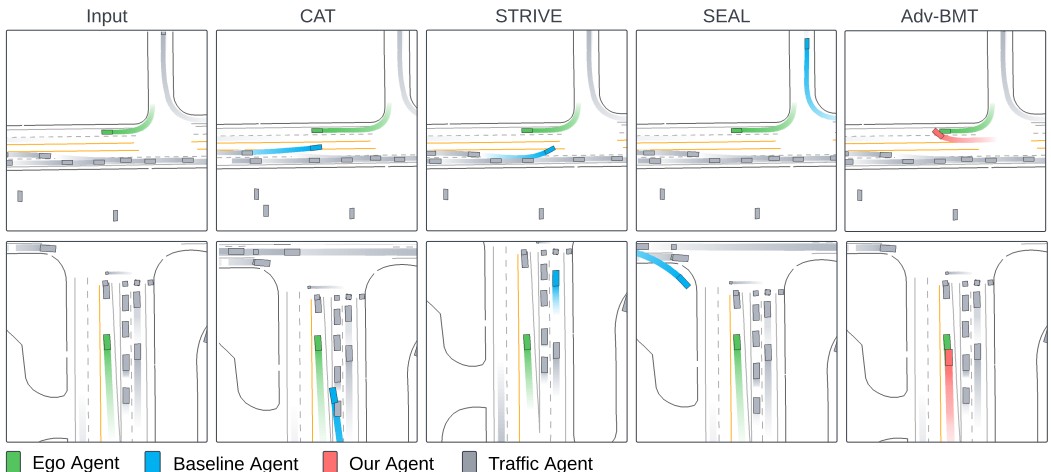

Ego Agent    Baseline Agent    Our Agent    Traffic Agent

Figure 7: **Baseline method comparisons.** We present several examples where Adv-BMT successfully generates collision interactions in scenarios where all baseline methods fail. Unlike the baselines, Adv-BMT does not rely on the proximity of neighboring agents to the ego vehicle, enabling more flexible and diverse adversarial attack strategies.

## 5   Conclusion

We introduced Adv-BMT, a novel framework for generating diverse safety-critical driving scenarios with realistic traffic interactions. Built on a Bidirectional Motion Transformer with the ability to perform bidirectional motion prediction tasks, Adv-BMT is able to design a new adversarial agent for each real-world scenario. A key advantage of Adv-BMT lies in its ability to generate realistic and flexible multi-agent interactions surrounding the adversarial agent. Unlike previous frameworks that select an existing neighboring vehicle and modify the corresponding trajectory for an adversarial attack, Adv-BMT initializes candidate collisions for a new adversarial agent, reconstructs multi-agent trajectories via inverse prediction, and optionally applies forward refinement for reactive, traffic-consistent interactions. Adv-BMT is able to balance realism, controllability, and scene reactivity. Our evaluations validate the quality of Adv-BMT scenarios in terms of interaction realism and diversity; furthermore, we demonstrate that learning within the Adv-BMT traffic flow improves AD agents' safety performance by a clear margin.

**Limitations.**   Adv-BMT relies on long token sequences to represent multi-agent traffic scenes, which results in high memory and computational demands. Our empirical study focuses on a single downstream task—reinforcement learning for an autonomous driving agent—so broader applicability remains to be validated.

**Acknowledgment**: This work was supported by the NSF Grants CNS-2235012, IIS-2339769, and CCF-2344955. ZP was supported by the Amazon Fellowship via UCLA Science Hub.

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

# A   Model Details

## A.1   BMT Architecture

**Preprocessing.**   The model takes real-world scenarios in the ScenarioNet format [8]. It performs a series of preprocessing steps on both static map features and dynamic agent trajectories. We compute the global bounds to extract a consistent map center and heading. Each map feature is decomposed into a sequence of vectorized segments, where each vector is represented by start and end points in 3D coordinates. These vectors are augmented with relative directional positions, segment headings, and lengths. Semantic attributes are binary indicators that encode feature types. Map feature types (e.g., lanes, crosswalks, lane markings, stop signs, etc.) are encoded as semantic indicators. Features are then centered at the map center. Traffic-light states are also encoded and aligned with timesteps. Similarly, agent trajectories are centered for agent-feature encoding, and position, heading, velocity, shape, and type are extracted over time to form temporal sequences. We extract a 16-dimensional state vector at each timestep. This preprocessing ensures that all trajectories are represented in a consistent spatial frame and expressed in an egocentric coordinate system.

**BMT Motion Tokenization.**   We formulate motion prediction as an autoregressive next-token prediction problem, where each motion token corresponds to a discretized control input over a fixed time interval. The tokenization process maps continuous agent motion—longitudinal acceleration and yaw rate—into discrete 2D bins. During tokenization, we consider candidate tokens sampled from a fixed grid; for each candidate, we simulate the resulting motion over a short duration and select the best-matching action by minimizing the contour-alignment error between the predicted agent footprint and the ground-truth pose (position and heading). We adopt a simplified bicycle model to parameterize agent motion using longitudinal acceleration and yaw rate within predefined bounds: acceleration $\in [-10, 10]\,\mathrm{m/s^2}$ and yaw rate $\in [-\pi/2, \pi/2]\,\mathrm{rad/s}$. Given predicted motion-token sequences, the trajectory is reconstructed by mapping tokens back to acceleration and yaw-rate changes.

In both forward and reverse directions, the same tokenizer is used to decode motion tokens into continuous trajectories. Given an initial state $\tau_t = (x_t, y_t, \theta_t, v_t)$, forward decoding simulates the next state $\tau_{t+1}$ by applying a tokenized control action $z_t = (a_t, \delta_t) \in \mathcal{A}$, where $\mathcal{A}$ is the discrete token space; repeating this autoregressively over the predicted tokens reconstructs the full trajectory $\boldsymbol{\tau}_{t:t+T}$. For reverse decoding, given a known future state $\tau_{t+1}$, the model evaluates candidate tokens $z_t \in \mathcal{A}$, simulates their inverse dynamics with $\Delta t \to -\Delta t$, and selects the token that best reconstructs the preceding state $\tau_t$. Operating in both directions is a key distinction of our approach: forward prediction enables open-loop rollouts of future behaviors, whereas reverse prediction traces from a desired outcome (e.g., a collision state) back to plausible initiating actions.

**BMT Motion Decoder.**   The decoder follows a GPT-like structure composed of stacked cross-attention blocks. Each block integrates three structured attention modules: agent-to-agent (A2A), agent-to-temporal (A2T), and agent-to-scene (A2S). Relational information across modalities is encoded via relation embeddings: agent-to-agent relation embedding, agent-to-time relation embedding, and agent-to-scene relation embedding, each capturing context-specific spatial/temporal cues. For token construction, we use several embeddings, including the agent-type embedding, agent-shape embedding, agent-ID embedding, and motion-token embedding (which embeds discrete control tokens). A continuous-motion feature embedding encodes acceleration and yaw-rate attributes. Auxiliary embeddings include the special-token embedding (to indicate sequence boundaries) and the reverse-prediction indicator embedding (to distinguish forward from reverse prediction modes). For each attention edge, a relative-relation embedding is computed using a Fourier encoder [20] and added to the key and/or value vectors. The input agent tokens are progressively updated across layers by aggregating contextual features from other agents, their temporal histories, and relevant map elements.

## A.2 BMT Motion Decoding

**Input.** Specifically, each input agent token to the BMT motion decoder is constructed by summing of embeddings of:

 a. the motion token from the previous step (representing discretized acceleration and yaw rate),

 b. agent shape (length, width, height),

 c. agent type (e.g., vehicle, pedestrian),

 d. agent identifier (embedded optionally),

 e. special token type (e.g., `<start>`, `<end>`, or padding),

as well as a continuous motion delta feature embedded via a Fourier encoder. These components are projected into the same hidden dimension and summed to form the input motion token embedding.

**Prediction Head and Output.** After processing through all decoding layers, the final hidden state for each valid token is passed through a two-layer MLP head to produce logits over the motion token space:

$$\text{MLP}(\mathbf{h}) = W_2 \cdot \phi(W_1 \cdot \mathbf{h}) \in \mathbb{R}^{K^2},$$

where $\phi(\cdot)$ denotes the GELU activation and $K^2$ is the number of discrete motion tokens (from a $K \times K$ acceleration–yaw bin grid). The resulting logits are used to predict the next motion token at each time step. During inference, we generate motion tokens using nucleus (top-$p$) sampling.

**Trajectory Reconstruction** Our model makes a prediction in the interval of 0.5 seconds. To simulate the effect of a motion token over a fixed time step $\Delta t = 0.5\,\text{s}$, we adopt midpoint integration based on a simplified bicycle model. In forward predictions, given a current state $\mathbf{s}_t = (x_t, y_t, \theta_t, v_t)$, the model computes the next speed and heading as $v_{t+1} = v_t + a \cdot \Delta t$ and $\theta_{t+1} = \theta_t + \omega \cdot \Delta t$. The average speed and heading are then defined as $\bar{v} = \frac{v_t + v_{t+1}}{2}$ and $\bar{\theta} = \left( \frac{\theta_t + \theta_{t+1}}{2} \right)$. In the reverse direction, the process is inverted. Given a future state $\mathbf{s}_{t+1}$, the model enumerates all possible token candidates and inverts the dynamics: $v_t = v_{t+1} - a \cdot \Delta t$ and $\theta_t = \theta_{t+1} - \omega \cdot \Delta t$. The average quantities $\bar{v}$ and $\bar{\theta}$ are computed similarly and used to derive the previous position:

$$x_t = x_{t+1} - \bar{v} \cdot \cos(\bar{\theta}) \cdot \Delta t, \quad y_t = y_{t+1} - \bar{v} \cdot \sin(\bar{\theta}) \cdot \Delta t.$$

## A.3 BMT Training

**Training Loss.** The decoder produces a logit tensor $\hat{\mathbf{z}} \in \mathbb{R}^{B \times T \times N \times |\mathcal{A}|}$, where $|\mathcal{A}|$ is the number of motion tokens (i.e., discretized acceleration–yaw pairs). The supervision target is the ground-truth token sequence $\mathbf{z}^* \in \mathbb{N}^{B \times T \times N}$, derived by tokenizing agent trajectories. A binary mask $\mathbf{m} \in \{0,1\}^{B \times T \times N}$ specifies which tokens are valid and should contribute to the training loss. The training objective is computed over all valid entries using the cross-entropy loss:

$$\mathcal{L}_{\text{main}} = \frac{1}{\sum_{b,t,n} m_{b,t,n}} \sum_{b,t,n} m_{b,t,n} \cdot \text{CE}(\hat{z}_{b,t,n}, z^*_{b,t,n}),$$

where CE denotes the standard cross-entropy loss between the predicted logits and the ground-truth discrete token.

**Reverse Prediction.** During reverse prediction, the model measures metrics separately for forward and reverse token predictions. Let $\mathbf{b} \in \{0,1\}^{B \times T \times N}$ be a binary indicator for whether each token comes from reverse prediction. Then we compute separate metrics:

$$\text{Accuracy}^{\text{reverse}} = \frac{\sum m_{b,t,n} \cdot b_{b,t,n} \cdot \mathbf{1}[\hat{z}_{b,t,n} = z^*_{b,t,n}]}{\sum m_{b,t,n} \cdot b_{b,t,n}},$$

$$\text{Entropy}^{\text{reverse}} = \frac{1}{\sum m \cdot b} \sum m_{b,t,n} \cdot b_{b,t,n} \cdot \mathcal{H}(\hat{z}_{b,t,n}),$$

with analogous expressions for forward prediction (i.e., for $1 - b_{b,t,n}$).

Table 6: BMT Model Parameters.

| Component | Parameters | Size (MB) |
|---|---|---|
| Scene Encoder | 902,080 | 3.44 |
|     Map Polyline Encoder | 22,656 | 0.09 |
|     Traffic Light Embedding MLP | 1,024 | 0.00 |
|     Scene Relation Embedding | 117,184 | 0.45 |
|     Scene Transformer Encoder | 744,448 | 2.84 |
|     Scene Encoder Output Projection | 16,512 | 0.06 |
|     Scene Output Pre-Normalization | 256 | 0.00 |
| Motion Decoder | 4,385,025 | 16.73 |
|     Multi-Cross Attention Decoder | 2,881,536 | 10.99 |
|     Motion Prediction Head | 157,121 | 0.60 |
|     Motion Prediction Pre-Normalization | 256 | 0.00 |
|     Agent-to-Agent Relation Embedding | 418,432 | 1.60 |
|     Agent-to-Time Relation Embedding | 418,432 | 1.60 |
|     Agent-to-Scene Relation Embedding | 117,184 | 0.45 |
|     Agent Type Embedding | 640 | 0.00 |
|     Motion Token Embedding | 139,520 | 0.53 |
|     Agent Shape Embedding | 17,152 | 0.07 |
|     Agent ID Embedding | 16,384 | 0.06 |
|     Continuous Motion Feature Embedding | 217,600 | 0.83 |
|     Special Token Embedding | 512 | 0.00 |
|     Reverse Prediction Indicator Embedding | 256 | 0.00 |
| Total | 5,287,105 | 20.17 |

**Metrics.** To measure the quality and diversity of the model's predictions during training, we track the perplexity:

$$\text{Perplexity} = \exp\left(-\sum_{a \in \mathcal{A}} \bar{p}_a \log(\bar{p}_a + \epsilon)\right), \quad \text{where} \quad \bar{p}_a = \frac{1}{M} \sum_{i=1}^{M} \mathbf{1}[\hat{z}_i = a],$$

and $M$ is the number of valid tokens. We also track the number of distinct tokens used by both predictions and ground truth:

$$\text{Cluster} = \sum_{a \in \mathcal{A}} \mathbf{1}[\bar{p}_a > 0].$$

**Total Loss.** The total loss is the sum of all enabled components:

$$\mathcal{L}_{\text{total}} = \mathcal{L}_{\text{main}} + \lambda_{\text{map}} \mathcal{L}_{\text{map}} + \lambda_{\text{tg}} \mathcal{L}_{\text{tg-total}},$$

with default weights $\lambda_{\text{map}} = \lambda_{\text{tg}} = 1$.

## A.4 Training Details

Our model has 5.2 million trainable parameters, with details indicated in Table 6. We trained BMT model on the training set of the Waymo Open Motion Dataset [4]. WOMD contains 480K real-world traffic with each scenario of length 9 seconds; traffic are composed by agents of vehicle, pedestrian, and bicycle; Each scenario comes with a high-fidelity road map. During training, we use 8 NVIDIA RTX A6000 GPUs for our model training and fine-tunings. We trained BMT in two stages, each with hyper-parameters indicated in Table 7. In the first stage, we pre-trained BMT for forward prediction only with 1 million steps. Then, we fine-tuned BMT with reverse motion prediction in fine-tuning with totally 1.5 million steps. We use AdamW optimizer for learning rate scheduling.

Table 7: BMT Training settings.

| Forward Prediction | | | Reverse Prediction | |
|---|---|---|---|---|
| Hyper-parameter | Value | | Hyper-parameter | Value |
| Training steps | 10E6 | | Training steps | 15E6 |
| Batch sizes | 2 | | Batch size | 2 |
| Training Time (h) | 185 | | Training Time (h) | 310 |
| Sampling Topp | 0.95 | | Sampling Topp | 0.95 |
| Sampling temperature | 1.0 | | Sampling temperature | 1.0 |
| Learning Rates | 3E-4 | | Learning Rates | 3E-4 |

Figure 8: Diverse adversarial behaviors generated by Adv-BMT.

# B  Additional Experiment Results

**Training Environment.**  We conduct our reinforcement learning experiments using the MetaDrive ScenarioEnv [9], which provides standardized driving environments for training and evaluating autonomous agents. Each environment encodes sensor observations including LiDAR-based surroundings and physical dynamics. Specifically, the observation space consists of three key components: (i) Ego state, which contains the ego vehicle's current physical state such as speed, heading, and steering; and (ii) surroundings, which encodes nearby traffic objects.

Actions are continuous and correspond to low-level vehicle control commands. The agent outputs a 2D normalized vector, which is then mapped to steering angle, throttle (acceleration), and brake signal. The environment includes a compositional reward structure combining driving progress, collision

Table 8: RL training settings.

| Adv-BMT | | | TD3 | |
| --- | --- | --- | --- | --- |
| Hyper-parameter | Value | | Hyper-parameter | Value |
| Scenario Horizon | 9s | | Discounted Factor | 0.99 |
| History Horizon | 0s | | Train Batch Size | 1024 |
| Collision Step | 1s–9s | | Learning Rate | 1E-4 |
| Prediction Mode | 8 | | Policy Delay | 200 |
| Policy Training Steps | 10E6 | | Target Network | 0.005 |

Table 9: Realism results for BMT's reverse prediction.

| Method | $\text{SFDE}_{avg}$ | $\text{SFDE}_{min}$ | $\text{SADE}_{avg}$ | $\text{SADE}_{min}$ | $\text{Coll}_{agent,min}$ | $\text{Coll}_{agent,avg}$ |
| --- | --- | --- | --- | --- | --- | --- |
| Reverse | **3.92** | **2.69** | **5.53** | **5.11** | **0.03** | **0.05** |
| Reverse + Adv-init | 7.17 | 5.56 | 7.68 | 7.12 | 0.13 | 0.16 |
| Reverse + Adv-init + Filter | 6.93 | 5.30 | 7.58 | 7.02 | 0.12 | 0.15 |

Table 10: Diversity results for BMT's reverse prediction.

| Method | FDD | ADD | $\text{JSD}_{vel}$ | $\text{JSD}_{acc}$ | $\text{JSD}_{TTC}$ |
| --- | --- | --- | --- | --- | --- |
| Reverse | 8.35 | 3.13 | **0.17** | 0.71 | **0.02** |
| Reverse + Adv-init | 9.89 | 3.82 | 0.22 | **0.64** | 0.13 |
| Reverse + Adv-init + Filter | **10.27** | **3.99** | 0.22 | 0.73 | 0.09 |

penalties, and road boundary violations. Driving reward is measured by forward lane progress, while penalties are applied for collisions with other vehicles or drifting off-road.

**Hyperparameter.** The settings of our open-loop and closed-loop adversarial RL experiments are shown in table 8. Note that in our closed-loop learning, Adv-BMT takes one frame of agent information as input for adversarial generations, whereas all baseline methods take one second agent history.

## C   Quantitative Results

### C.1   Ablation Study

In Table 9 and Table 10, reverse predictions with adversarial initializations exhibit greater deviation from the ground-truth data and yield improved diversity. This behavior is expected, since adversarial initializations modify the terminal positions and headings of selected agents, which propagates backward into more varied histories. Importantly, our rule-based filtering does not reduce diversity; rather, it preserves multimodality while improving realism metrics by removing physically implausible trajectories.

### C.2   Waymo Open Sim Agents Challenge

We evaluate BMT results on 400 WOMD validation scenarios using the Waymo Open Sim Agents Challenge (WOSAC) 2025 [11]. Evaluation results are summarized in Table 11. For the metrics, smaller values of minADE indicate more accurate predictions, whereas larger values for the remaining metrics indicate better performance. From the results, we observe that forward prediction achieves much better performance than reverse prediction across all metrics, except for similar performance in angular speed, angular acceleration, distance to the nearest object, and TTC. Note that the training times for forward prediction and reverse prediction are similar (10E6 and 15E6 steps, respectively). The WOSAC results indicate that our BMT model is better at predicting future motion than historical motion.

Table 11: WOSAC Evaluation results of BMT.

| Metrics | Reverse | Forward |
|---|---|---|
| Linear speed | 0.375 | 0.393 |
| Linear acceleration | 0.394 | 0.405 |
| Angular speed | 0.441 | 0.428 |
| Angular acceleration | 0.594 | 0.593 |
| Distance to nearest object | 0.405 | 0.388 |
| Collision | 0.521 | 0.951 |
| Time to Collide | 0.840 | 0.840 |
| Distance to road edge | 0.675 | 0.683 |
| Offroad | 0.564 | 0.934 |
| Realism | 0.554 | 0.753 |
| Kinematic | 0.451 | 0.455 |
| Interactive | 0.566 | 0.801 |
| Map | 0.596 | 0.862 |
| minADE | 2.148 | 1.344 |
| Metametric | 0.554 | 0.753 |

Upon reviewing the collisions detected in reverse predictions with real initializations, we found that most occurred in crowded parking-lot areas, where clusters of parked vehicles or pedestrians are close together. These were flagged as collisions by WOSAC's evaluation API, even though they may not represent meaningful agent–agent interactions. This explains why metrics such as ADE and FDE remain comparable across methods despite differences in collision scores.

The performance gap is primarily due to our training strategy: the model was pre-trained for forward prediction (around 800k steps) and then fine-tuned for bidirectional prediction (around 1.5M steps). We have included these details in Section A.4 (Training Details).

# D Qualitative Results

**Visualizations.** Figure 8 and Figure 9 present a set of example scenarios results generated by Adv-BMT. Across different scenarios, the adversarial agent exhibits a diverse range of safety-critical driving behaviors, demonstrating their ability to interact plausibly with realistic traffic participants. The visualizations illustrate that Adv-BMT can generate multiple distinct collision outcomes from a single driving log. This highlights a key advantage of Adv-BMT over baseline methods, which tend to produce identical or highly similar adversarial behaviors for the same input scenario.

**Demo Video.** We submit a video within our supplementary materials. Here we provide visualizations with case studies through animated simulations of Adv-BMT scenarios, which include different types of vehicles, pedestrians, and bicycle agents. More demos are available at https://metadriverse.github.io/adv-bmt/.

# E Broader Impacts

Our work introduces a novel model for generating safety-critical traffic scenarios, aiming to improve the safety reliability and driving robustness of AD systems. By modeling both forward and reverse motion trajectories, our framework enables controllable and diverse simulation of rare and high-risk traffic events. Our framework, Adv-BMT, benefits the development and testing of safer autonomous agents by exposing failure cases under challenging interactions. However, generating adversarial scenarios may potentially raise concerns about potential misuse, such as crafting unrealistic or malicious simulations. To address this, our approach is designed for research and evaluation within closed simulation environments. We encourage responsible usages of our model and encourage integrating them into safety validation pipelines with appropriate regulations.

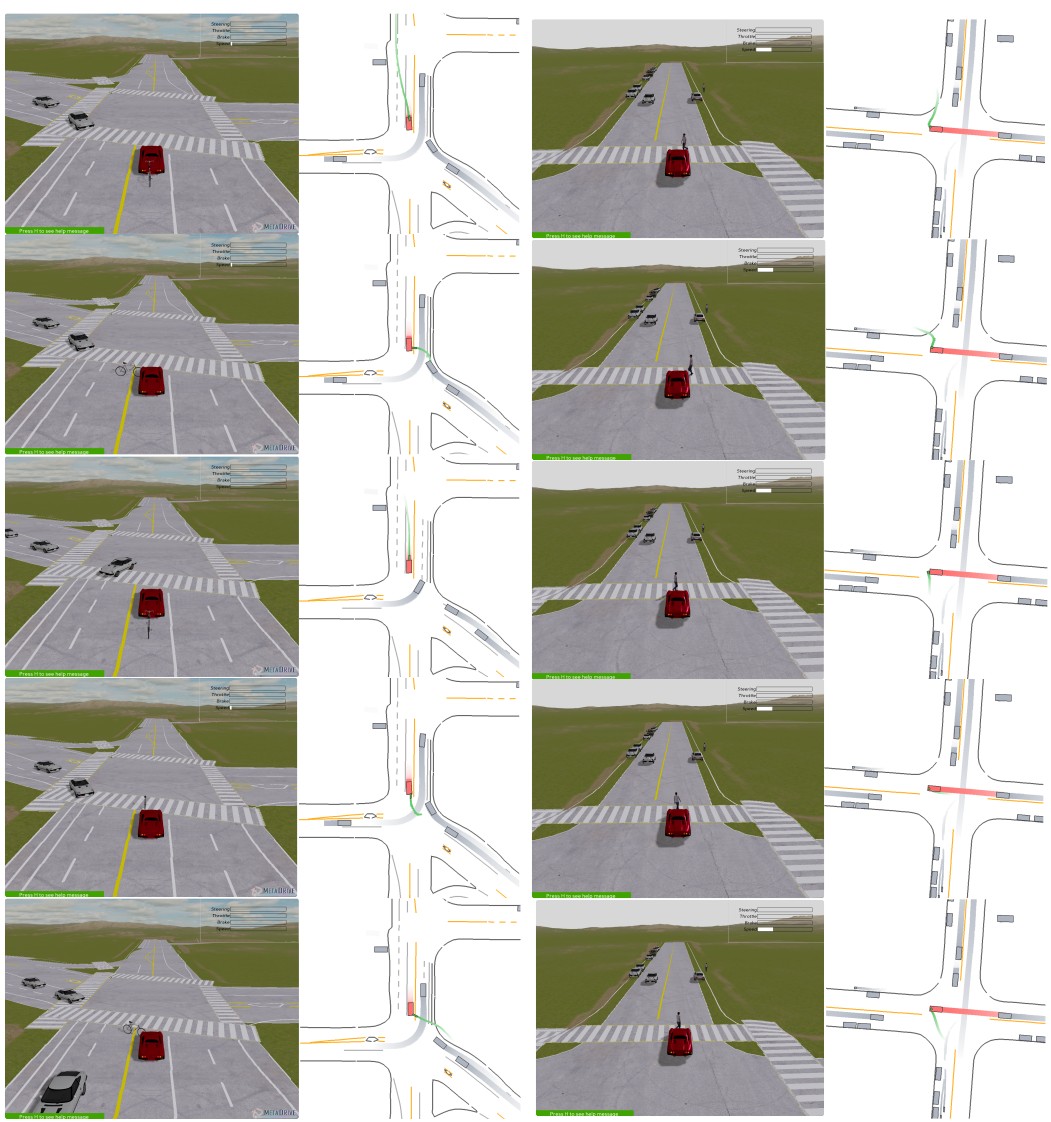

Figure 9: Diverse adversarial behaviors generated by Adv-BMT.

