# OpenReview forum: "Adv-BMT: Bidirectional Motion Transformer for Safety-Critical Traffic Scenario Generation"
_NeurIPS.cc/2025/Conference — NeurIPS 2025 poster_

### Official Review · Reviewer_v8s5 · 2025-06-08

**Clarity:** 3
**Significance:** 2
**Originality:** 2
**Rating:** 4
**Confidence:** 4

**Summary:**

This paper introduces a new method for generating realistic and adversarial scenarios by reverse prediction from a collision scenario. They demonstrate that scenarios generated with Adv-BMT improve robustness and safety of downstream policies, surpassing several prior adversarial scenario generation approaches.

**Questions:**

1. Address the weakness.
2. Why is the improvement from training the agent using RL with the scenarios generated with the BMT method so limited?
3. Why does predicting history motion achive much worse realism than predicting future motion?

**Ethical Concerns:**

["NO or VERY MINOR ethics concerns only"]

**Final Justification:**

The additional experiments and clarifications have addressed my concerns.

**Limitations:**

Yes.

**Quality:**

3

**Strengths And Weaknesses:**

Strengths:

1. Generating the collision scenario resersely is novel.
2. The paper is well-writen and figure is clear.
3. The experiment is extensive with real world data.

Weakness:
1. The realism of the reversely generated scenario is questionable. The result in table 2 demonstrates the baseline can achive better realism. Table 8 also demonstrates the reservely generated scenarios have a poor WOSAC realism meta metric.
2. The method uses rule-based filter but it is not applied to the baseline method. So the comparison is unfair.
3. In figure 4, different method uses different adversarial agents. Consider using the same adversarial agent and explain why the trajectory of the baseline method is so weird.

---

> ### Author Rebuttal · Authors · 2025-07-31
>
> We thank the reviewer for all valuable questions and suggestions.
>
> > 1. The realism of the reversely generated scenario is questionable. The result in table 2 demonstrates the baseline can achive better realism. Table 8 also demonstrates the reservely generated scenarios have a poor WOSAC realism meta metric.
>
> We appreciate the reviewer’s concern regarding the realism of the reversely generated scenarios. As shown in Table 8, reverse prediction results exhibit lower scores in collision likelihood and offroad likelihood, which contributes to a lower WOSAC realism meta-metric compared to forward predictions. The reverse model is designed to predict the behavior of an adversarial agent, which allows predicted collisions and deviates from typical driving patterns.
> More importantly, our ablation study table attached below shows that the reverse model produces adversarial behaviors with both higher realism and diversity than baseline generation strategies that do not incorporate learned backward agent dynamics. Rule-based filters improve generation quality for the adversarial agents and remove implausible or extreme behaviors, enhancing the realism of the final scenarios.
> We leverage the high performance of forward predictions in Adv-BMT’s forward refinement step to allow realistic interactions among traffic agents. As supported by both our quantitative metrics and qualitative visualizations, the resulting scenarios are realistic, diverse, and safety-critical, serving as effective test cases for evaluating prediction models.
>
> >2. The method uses rule-based filter but it is not applied to the baseline method. So the comparison is unfair.
>
> When attempting to apply these filters to baseline methods such as CAT, SEAL and STRIVE, we realize that the adversarial behaviors produced by the implementations lack the variability needed to generate alternative trajectories once the initial ones were rejected by the filters. This is also indicated by the low FDD and SDD scores in our evaluation. So we update the ablation study and remove filtering from Adv-BMT in Table 2.
>
> a. Realism of generated Adversarial behavior
>
> | Method              | Adversarial Attack Success | avg SFDE | min SFDE | avg SADE | min SADE | min Agent Collision | avg Agent Collision | min ADV-Traffic Collision | avg ADV-Traffic Collision |
> |---------------------|----------------------------|----------|----------|----------|----------|---------------------|---------------------|---------------------------|---------------------------|
> | CAT                 | 0.48                       | 1.01     | 1.01     | 0.31     | 0.31     | 0.1                 | 0.1                 | 0.3                       | 0.3                       |
> | SEAL                | 0.55                       | 1.26     | 1.26     | 0.38     | 0.38     | 0.12                | 0.12                | 0.32                      | 0.32                      |
> | STRIVE              | 0.38                       | 0.63     | 0.63     | 0.17     | 0.17     | 0.09                | 0.09                | 0.32                      | 0.32                      |
> | BMT_TF              | 1                          | 2.27     | 1.67     | 1.51     | 1.3      | 0.13                | 0.15                | 0.22                      | 0.39                      |
> | BMT_All             | 1                          | 2.36     | 1.63     | 1.56     | 1.29     | 0.13                | 0.15                | 0.19                      | 0.37                      |
> |                     |                            |          |          |          |          |                     |                     |                           |                           |
>
> (Note: here the min/avg meaning the mode selection for each predicted scenario)
>
> b. Diversity of generated Adversarial behavior
> | Method              | Final Displacement Diversity | Average Displacement Diversity | Velocity JSD | Acceleration JSD | Time-To-Collide JSD |
> |---------------------|------------------------------|--------------------------------|--------------|------------------|---------------------|
> | CAT                 | 0                            | 0                              | 0.13         | 0.16             | 0.12                |
> | SEAL                | 0                            | 0                              | 0.18         | 0.25             | 0.17                |
> | STRIVE              | 0.01                         | 0                              | 0.11         | 0.22             | 0.12                |
> | BMT_TF              | 1.22                         | 0.49                           | 0.1          | 0.17             | 0.13                |
> | BMT_All             | 1.51                         | 0.63                           | 0.1          | 0.17             | 0.08                |
>
>
> >3. In figure 4, different method uses different adversarial agents. Consider using the same adversarial agent and explain why the trajectory of the baseline method is so weird.
>
> We will update the figure to use the same initial adversarial agent across all methods for consistency, and revise the caption to explain any trajectory artifacts observed in the baseline methods.
>
>
> >4. Why is the improvement from training the agent using RL with the scenarios generated with the BMT method so limited?
>
> We appreciate the reviewer’s question regarding the impact of using Adv-BMT-generated scenarios for reinforcement learning. As shown in Table 3 (open-loop RL) and Table 4 (closed-loop RL), agents trained on Adv-BMT scenarios consistently achieved lower collision rates and reduced trajectory costs compared to baseline methods. Specifically, in the open-loop setting, our agent outperformed the best baseline by 5% in average cost, 8% in collision rate, and 10% in average trajectory cost. In the closed-loop setting, we observed improvements of 3% in average cost, 10% in collision rate, and 38% in average trajectory cost. This highlights the value of the diverse and targeted adversarial scenarios generated by Adv-BMT in strengthening policy robustness through reinforcement learning.
>
>
> >5. Why does predicting history motion achive much worse realism than predicting future motion?
> We attach our WOSAC results below:
>
> a. Forward prediction:
>
> | Metric name                             | Value       |
> |----------------------------------------|-------------|
> | Realism meta-metric                    | 0.7694586   |
> | Linear Speed Likelihood                | 0.37768704  |
> | Linear Acceleration Likelihood         | 0.39826843  |
> | Angular Speed Likelihood               | 0.42033565  |
> | Angular Acceleration Likelihood        | 0.59226924  |
> | Distance To Nearest Object Likelihood  | 0.38673738  |
> | Collision Likelihood                   | 0.9600281   |
> | Time To Collision Likelihood           | 0.82912844  |
> | Distance To Road Edge Likelihood       | 0.6734454   |
> | Offroad Likelihood                     | 0.9424464   |
> | minADE                                 | 1.4130014   |
>
>
> b. Reverse prediction:
>
> | Metric Name                          | Value      |
> |-------------------------------------|------------|
> | Realism meta-metric                 | 0.5640852  |
> | Linear Speed Likelihood             | 0.36751205 |
> | Linear Acceleration Likelihood      | 0.38674814 |
> | Angular Speed Likelihood            | 0.43716192 |
> | Angular Acceleration Likelihood     | 0.59631693 |
> | Distance To Nearest Object Likelihood | 0.40195426 |
> | Collision Likelihood                | 0.5159998  |
> | Time To Collision Likelihood        | 0.8315656  |
> | Distance To Road Edge Likelihood    | 0.6667353  |
> | Offroad Likelihood                  | 0.5591505  |
> | minADE                              | 2.160784   |
>
> We appreciate the reviewer’s question regarding the lower realism of the reversely generated scenarios. As shown in Table 8, the reverse model yields a lower overall WOSAC realism meta-metric (0.56) compared to the forward model (0.77). This gap is primarily due to reduced scores in collision likelihood (0.52 vs. 0.96) and offroad likelihood (0.56 vs. 0.94). These two metrics heavily influence the realism meta-metric and suggest that reverse-predicted trajectories more frequently result in unsafe or abnormal behaviors, such as agent collisions or off-road motions. Despite this, the reverse model performs comparably to the forward model in some dynamics-related metrics, such as linear and angular acceleration likelihood, indicating that the generated motions are still kinematically plausible.

---

> > ### Comment · Reviewer_v8s5 · 2025-08-02
> > **More questions**
> >
> > Thanks for the rebutal. I still have some questions.
> >
> > 1. Is the acceleration and heading tokenization better than the k-dist tokenization in Smart?
> > 2. Will this work be open-source?
> > 3. Why reverse-predicted trajectories more frequently result in unsafe or abnormal behavior. Does the reverse prediction use an adversarial initialization or real initialization? If it uses adversarial initialization, what is the real initialization performance? If it uses real initialization, what is the adversarial initialization performance?

---

> ### Author Response · Authors · 2025-08-03
> **Reply to the follow-up questions**
>
> Thank you for your response to our rebuttal.
>
> > 1. Is the acceleration and heading tokenization better than the k-dist tokenization in Smart?
>
> A major reason to stick with acceleration and heading tokenization is the bidirectional symmetry. For k-disk tokenization, which clusters short trajectory segments based on motion patterns, lacks this symmetry. One would need to reconstruct a separate motion vocabulary tailored for backward motion. Besides, in the original SMART paper, the author concluded that ‘The k-disks approach loses fine-grained trajectory information during discretization, but good for zero-shot generalizations’, so k-dist is not our preferred choice.
>
> Empirically for forward prediction, we also observed that k-disk tokenization yields a higher reconstruction error, measured by the average $\ell_2$ distance between predicted and ground-truth bounding box corners, compared to the acceleration and heading approach. However, one trade-off is that our method results in slightly lower rotational accuracy under WOSAC evaluation. This is due to the need to interpolate heading values at intermediate timestamps (e.g., at 0.1s, 0.2s, etc.), since our tokens predict acceleration and heading changes over coarser 0.5s intervals. While this interpolation introduces minor noise in rotational dynamics, we find the benefits in bidirectional compatibility and reconstruction accuracy outweigh this limitation for our use case.
>
>
>
> >2. Will this work be open-source?
>
> Yes, we will make the code and model checkpoint public in September, aligned with the final paper decision and after incorporating necessary revisions.
>
>
> >3. Why reverse-predicted trajectories more frequently result in unsafe or abnormal behavior. Does the reverse prediction use an adversarial initialization or real initialization? If it uses adversarial initialization, what is the real initialization performance? If it uses real initialization, what is the adversarial initialization performance?
>
> Thanks a lot for the valuable insight questions!
>
> 1. In the WOSAC evaluation, we used real initializations only for reverse prediction. We have now included an ablation study comparing both real and adversarial initializations for completeness.
>
>
> 2. Upon reviewing the collisions detected in reverse predictions with real initializations, we found that most occurred in crowded parking lot areas, where clusters of parked vehicles or pedestrians are close together. These are flagged as collisions. This explains why metrics such as ADEs and FDEs remain comparable across methods despite differences in collision scores. The performance gap between forward and reverse prediction is due to our training strategy: the model was pre-trained for forward prediction (around 800k steps) and then fine-tuned for bidirectional prediction (around 1.5M steps). We have included these details in Section A.2 (Training Details) of the appendix, and we will clarify this point further in the revised manuscript.
>
> a. Realism
> | Method                      | avg SFDE | min SFDE | avg SADE | min SADE | min Agent Collision Rate | avg Agent Collision Rate |
> |-----------------------------|----------|----------|----------|----------|--------------------------|--------------------------|
> | Reverse                     | 3.92     | 2.69     | 5.53     | 5.11     | 0.03                     | 0.05                     |
> | Reverse + Adv-init          | 7.17     | 5.56     | 7.68     | 7.12     | 0.13                     | 0.16                     |
> | Reverse + Adv-init + Filter | 6.93     | 5.3      | 7.58     | 7.02     | 0.12                     | 0.15                     |
> | | | |
>
>
> b. Diversity
> | Method                      | Final Displacement Diversity | Average Displacement Diversity | Velocity JSD | Acceleration JSD | Time-To-Collide JSD |
> |-----------------------------|------------------------------|--------------------------------|--------------|------------------|---------------------|
> | Reverse                     | 8.35                         | 3.13                           | 0.17         | 0.71             | 0.02                |
> | Reverse + Adv-init          | 9.89                         | 3.82                           | 0.22         | 0.64             | 0.13                |
> | Reverse + Adv-init + Filter | 10.27                        | 3.99                           | 0.22         | 0.73             | 0.09                |
> | | | |
>
> **Analysis**: Reverse predictions with adversarial initializations would deviate more from the ground-truth data with better diversity. This is expected since the adversarial initializations would modify the last-step position and heading for the selected adversarial agent.

---

> ### Author Response · Authors · 2025-08-04
>
> We greatly appreciate the time and effort you dedicated to evaluating our work. If you have any remaining comments, concerns or suggestions, we would sincerely appreciate the opportunity to consider them and incorporate your feedback into revisions.
>
> Thank you again for your contribution to the review process and for your service to the community.

---

### Official Review · Reviewer_qsAw · 2025-06-17

**Clarity:** 2
**Significance:** 3
**Originality:** 3
**Rating:** 4
**Confidence:** 3

**Summary:**

Summary

This paper addresses the challenge of limited coverage of long-tailed, safety-critical scenarios in existing real-world autonomous driving (AD) datasets, which hampers effective scenario-based validation. To overcome this limitation, the authors propose Adv-BMT, a framework designed to augment real-world scenarios with realistic and diverse adversarial interactions.
At the core of the method is a Bidirectional Motion Transformer (BMT) model that performs inverse traffic motion prediction. Given the final frame of a scenario, BMT reconstructs the preceding traffic interactions in reverse chronological order, back to the initial state. The framework follows a two-stage pipeline: it first performs adversarial initialization, then applies the inverse motion prediction to synthesize plausible collision scenarios. Unlike prior work, Adv-BMT does not require access to real-world collision data for pretraining, yet is still capable of generating realistic and diverse collision events. Experiments demonstrate that training with the augmented scenarios produced by Adv-BMT reduces the episode-level collision rate by 20% compared to previous methods. The authors also state their intention to release the code.

**Questions:**

1. The bidirectional prediction process in the paper is not clearly explained. Is it correct that the model first predicts from the last frame to the first (reverse), and then from the first frame to the last (forward)? During the reverse process in Adv-BMT, it is stated that the ego and all non-ADV agents are teacher-forced — in the forward refinement stage, are the trajectories of the ADV agent and ego vehicle (from the reverse process) also teacher-forced? In Table 1, for the "Bidirection" setting, is it correct that no teacher-forcing is applied at all?

2. There are some concerns regarding completeness of the experimental comparison. Why is the generation speed not compared against CAT? Could you also include the Adv-BMT results in Table 2 for completeness?

3. The related work section is missing several relevant baselines. For example, there is no mention of SEAL, and diffusion-based trajectory generation methods are not discussed. In particular, works that apply diffusion with a fixed goal for trajectory denoising are highly relevant and deserve comparison or discussion.

4. Could the authors provide more detail about the rule-based filtering process? Specifically, how many trajectories were generated initially, and how many were filtered out? What is the performance impact if the rule-based filtering is removed (e.g., on Table 2 results)? Since Adv-BMT applies rule-based filtering while many competing methods do not, this raises fairness concerns in the evaluation.

5. You mention that you will release the code. Could the authors clarify the timeline and the scope of the release?

**Ethical Concerns:**

["NO or VERY MINOR ethics concerns only"]

**Final Justification:**

The additional experiments and clarifications have addressed part of my concerns. However, since some of the rebuttal content was submitted beyond the valid rebuttal period, I will maintain the current score in accordance with NeurIPS rules.

**Limitations:**

Yes

**Quality:**

3

**Strengths And Weaknesses:**

### **Strengths**

1. The paper introduces the Bidirectional Motion Transformer (BMT), which employs temporally reversible motion tokenization to enable inverse traffic motion prediction.

2. The proposed method can generate diverse safety-critical scenarios efficiently, demonstrating both high scenario diversity and fast generation speed.

---

### **Weaknesses**

1. The writing could be improved for clarity. For example:
   - Table 2 does not explain what "SDD" refers to.
   - Tables 3 and 4 could benefit from highlighting the second-best results for better comparison.
   - In Table 1, the number of frames corresponding to each context length should be clearly specified. Does "context 0" represent a single-frame input? How many frames are included in 1 second?

2. The overall novelty appears to be limited

---

> ### Author Rebuttal · Authors · 2025-07-31
>
> We appreciate the reviewer's suggestions and questions. We will address point by point.
>
>
> >1. The writing could be improved for clarity. For example: Table 2 does not explain what "SDD" refers to. Tables 3 and 4 could benefit from highlighting the second-best results for better comparison.
>
> Thank you for the helpful suggestions regarding clarity. We will revise the manuscript to include a clear definition of Start Displacement Diversity (SDD), which quantifies the variability in predicted starting positions across different modes. Additionally, we will update Tables 3 and 4 to highlight the second-best results, improving visual comparison across methods.
>
>
> >2. In Table 1, the number of frames corresponding to each context length should be clearly specified. Does "context 0" represent a single-frame input? How many frames are included in 1 second?
>
> Yes, thank you for the question. A 0-second context refers to the setting where BMT takes only a single-frame of agent motion as input. In contrast, a 1-second context corresponds to 10 input frames, based on a frame rate of 10 Hz. We will revise the manuscript to clarify this definition in Table 1 and the corresponding text.
>
> >3. The overall novelty appears to be limited.
>
> We want to emphasize the novelty of Adv-BMT compared to other baseline methods. Adv-BMT is designed with flexible generation modes that support varying levels of multi-agent interaction:
>
> **Adv-BMT with traffic agent teacher-forced:** In this generation mode, traffic agents are teacher-forced with their recorded trajectories. The new adversarial agent interacts with replayed traffic agents in the driving log to keep the high scene fidelity by preserving the original traffic flow.
>
> **Adv-BMT with closed-loop reverse prediction:** all agents including traffic participants and the adversarial agent are predicted jointly. It allows the entire scene to evolve coherently, enabling multi-agent interactions and producing fully synthesized scenarios with ego collisions.
>
> **Adv-BMT with forward refinement:** To introduce reactivity without sacrificing the structure of the recorded scene, this mode performs an additional forward-prediction step after generating the adversarial agent in teacher-forced mode. This allows background agents to adjust their behavior in response to the adversarial agent’s presence.
> These complementary modes offer a trade-off between scene realism, controllability, and interactivity, enabling users to select the appropriate setting based on their evaluation needs.
>
> We thank the reviewer for the insightful feedback and will clarify in the paper that Adv-BMT is designed to support a range of interaction modes, allowing users to balance realism and diversity according to their specific use cases.
>
> >4. The bidirectional prediction process in the paper is not clearly explained. Is it correct that the model first predicts from the last frame to the first (reverse), and then from the first frame to the last (forward)?
>
> Yes, that is correct. In the bidirectional prediction process, BMT first performs reverse prediction, generating the initial frame from the given final frame. Then, using the predicted initial frame for all agents, BMT performs forward prediction to reconstruct the full trajectory up to the final frame. We will revise the manuscript to clarify this process.
>
> >5. During the reverse process in Adv-BMT, it is stated that the ego and all non-ADV agents are teacher-forced — in the forward refinement stage, are the trajectories of the ADV agent and ego vehicle (from the reverse process) also teacher-forced?
>
> In the reverse pass, the ego and traffic agents are teacher-forced while the adversarial agent is predicted. In the subsequent forward pass, the ego and adversarial agents (i.e., the collision participants) are teacher-forced, and the traffic agents are predicted to allow for interaction with the adversarial behavior. We will clarify this bidirectional teacher-forcing setup in the manuscript.
>
> >5. In Table 1, for the "Bidirection" setting, is it correct that no teacher-forcing is applied at all?
>
> Yes, that is correct. In the "Bidirection" setting in Table 1, the forward pass uses the predicted initial states from the reverse pass without applying teacher-forcing to any agents. This setup allows us to evaluate the realism and consistency of fully generated trajectories under our bidirectional prediction pipeline. We will clarify this in the revised manuscript.
>
> Ablation Studies Evaluation Results:
> Realism
> | Method            | avg SFDE | min SFDE | avg SADE | min SADE | veh_coll_scenario_count | veh_coll_min | veh_coll_avg |
> |-------------------|----------|----------|----------|----------|-------------------------|--------------|--------------|
> | Reverse           | 3.92     | 2.7      | 5.53     | 5.11     | 0.7                     | 0.03         | 0.05         |
> | Forward           | 3.52     | 2.35     | 2.39     | 1.98     | 0.6                     | 0.03         | 0.05         |
> | Reverse + Forward | 3.29     | 2.01     | 2.38     | 1.74     | 0.73                    | 0.03         | 0.06         |
>
>
> Diversity
> | Method            | Start Displacement Diversity | Final Displacement Diversity | Average Displacement Diversity | Velocity JSD | Acceleration JSD | Time-To-Collide JSD |
> |-------------------|------------------------------|------------------------------|--------------------------------|--------------|------------------|---------------------|
> | Reverse           | 8.35                         | 0                            | 3.13                           | 0.17         | 0.71             | 0.02                |
> | Forward           | 0                            | 10.78                        | 4.4                            | 0.23         | 0.69             | 0.3                 |
> | Reverse + Forward | 8.49                         | 12.78                        | 7                              | 0.22         | 0.77             | 0.35                |
>
> **Analysis:** The results show that bidirectional prediction (Reverse + Forward) enhances the diversity of generated agent behaviors across all displacement diversity metrics. In addition, it improves the realism of predictions, as evidenced by lower SFDE and SADE values, with a slight regression in average agent collision rate compared to the reverse-only and forward-only settings. This supports the effectiveness of the forward-refinement design in Adv-BMT to improve the realism of surrounding traffic participants.
>
> >6. There are some concerns regarding completeness of the experimental comparison. Why is the generation speed not compared against CAT?
>
> Thank you for pointing this out. The omission of CAT’s generation speed in Table 2 was an oversight on our part. We will correct this in the revised version and include CAT’s generation speed for a complete and fair comparison.
> | Method  | CAT  | SEAL | STRIVE | Adv-BMT |
> |---------|------|------|--------|---------|
> | 500-avg | 0.8  | 2.36 | 9.53   | 1.02    |
> | 500-std | 0.09 | 0.41 | 0.28   | 0.02    |

---

> ### Author Response · Authors · 2025-08-03
>
> >7. Could you also include the Adv-BMT results in Table 2 for completeness? What is the performance impact if the rule-based filtering is removed (e.g., on Table 2 results)? Since Adv-BMT applies rule-based filtering while many competing methods do not, this raises fairness concerns in the evaluation.
>
> We updated the table accordingly:
>
> a. Realism of generated Adversarial behavior
> | Method              | Adversarial Attack Success | avg SFDE | min SFDE | avg SADE | min SADE | min Agent Collision | avg Agent Collision | min ADV-Traffic Collision | avg ADV-Traffic Collision |
> |---------------------|----------------------------|----------|----------|----------|----------|---------------------|---------------------|---------------------------|---------------------------|
> | CAT                 | 0.48                       | 1.01     | 1.01     | 0.31     | 0.31     | 0.1                 | 0.1                 | 0.3                       | 0.3                       |
> | SEAL                | 0.55                       | 1.26     | 1.26     | 0.38     | 0.38     | 0.12                | 0.12                | 0.32                      | 0.32                      |
> | STRIVE              | 0.38                       | 0.63     | 0.63     | 0.17     | 0.17     | 0.09                | 0.09                | 0.32                      | 0.32                      |
> | Adv-BMT_TF              | 1                          | 2.27     | 1.67     | 1.51     | 1.3      | 0.13                | 0.15                | 0.22                      | 0.39                      |
> | Adv-BMT_All             | 1                          | 2.36     | 1.63     | 1.56     | 1.29     | 0.13                | 0.15                | 0.19                      | 0.37                      |
> |
>
>
> b. Diversity of generated Adversarial behavior
> | Method              | Final Displacement Diversity | Average Displacement Diversity | Velocity JSD | Acceleration JSD | Time-To-Collide JSD |
> |---------------------|------------------------------|--------------------------------|--------------|------------------|---------------------|
> | CAT                 | 0                            | 0                              | 0.13         | 0.16             | 0.12                |
> | SEAL                | 0                            | 0                              | 0.18         | 0.25             | 0.17                |
> | STRIVE              | 0.01                         | 0                              | 0.11         | 0.22             | 0.12                |
> | BMT_TF              | 1.22                         | 0.49                           | 0.1          | 0.17             | 0.13                |
> | BMT_All             | 1.51                         | 0.63                           | 0.1          | 0.17             | 0.08                |
> |
>
> (Note: here the min/avg meaning the mode selection for each predicted scenario)
>
> **Analysis:** Adv-BMT methods outperform baselines in both diversity metrics and adversarial attack success rates. The highly diverse adversarial behaviors produced by Adv-BMT are accompanied by a moderate increase in average displacement errors compared to the ground truth. Importantly, the interaction realism score (measured by collisions between adversarial and traffic agents) is comparable to or even better than that of the baselines.
>
>
>
> >8. The related work section is missing several relevant baselines. For example, there is no mention of SEAL, and diffusion-based trajectory generation methods are not discussed. In particular, works that apply diffusion with a fixed goal for trajectory denoising are highly relevant and deserve comparison or discussion.
>
> We agree that SEAL is an important and relevant baseline, and we will update the related work section as part of diffusion-based trajectory generation methods, and clarify how our approach differs from the baseline.

---

> ### Author Response · Authors · 2025-08-03
>
> >9. Could the authors provide more detail about the rule-based filtering process? Specifically, how many trajectories were generated initially, and how many were filtered out?
>
> In Adv-BMT, adversarial trajectories are generated in batches of six using diverse collision initializations. If at least one trajectory in the batch satisfies the rule-based filtering criteria, it is selected; therwise Adv-BMT will re-initialize the collision and re-predicts the trajectories. This re-sampling process is repeated up to five times. On a test set of 100 evaluation scenarios from the Waymo dataset, we observed that approximately 61% of the initially generated adversarial trajectories were rejected by the filters, indicating that about more than half of the sampled initializations might not be realistic enough to meet our constraints and required resampling.
>
> a. Realism
> | Method              | Adversarial Attack Success | avg SFDE | min SFDE | avg SADE | min SADE | min Agent Collision | avg Agent Collision | min ADV-Traffic Collision | avg ADV-Traffic Collision |
> |---------------------|----------------------------|----------|----------|----------|----------|---------------------|---------------------|---------------------------|---------------------------|
> | BMT_TF              | 1                          | 2.27     | 1.67     | 1.51     | 1.3      | 0.13                | 0.15                | 0.22                      | 0.39                      |
> | BMT_All             | 1                          | 2.36     | 1.63     | 1.56     | 1.29     | 0.13                | 0.15                | 0.19                      | 0.37                      |
> | BMT_TF_with_Filter  | 1                          | 2.28     | 1.35     | 1.53     | 1.15     | 0.13                | 0.14                | 0.12                      | 0.33                      |
> | BMT_All_with_Filter | 1                          | 1.97     | 1.23     | 1.4      | 1.09     | 0.12                | 0.14                | 0.09                      | 0.33                      |
> | | | | | |
>
> b. Diversity
> | Method              | Final Displacement Diversity | Average Displacement Diversity | Velocity JSD | Acceleration JSD | Time-To-Collide JSD |
> |---------------------|------------------------------|--------------------------------|--------------|------------------|---------------------|
> | BMT_TF              | 1.22                         | 0.49                           | 0.1          | 0.17             | 0.13                |
> | BMT_All             | 1.51                         | 0.63                           | 0.1          | 0.17             | 0.08                |
> | BMT_TF_with_Filter  | 2.32                         | 1                              | 0.08         | 0.12             | 0.09                |
> | BMT_All_with_Filter | 1.97                         | 0.83                           | 0.13         | 0.14             | 0.08                |
> | | | | | |
>
> **Analysis:** Introducing rule-based filters refines quality of adversarial trajectories. There are improvements in all realism-related metrics, with small shifts on diversity metrics. This affirms that our filtering module benefits prediction realism for Adv-BMT without compromising the behavioral diversities.
>
> >10. You mention that you will release the code. Could the authors clarify the timeline and the scope of the release?
>
> We plan to release the code around September, aligned with the final paper decision and after incorporating necessary revisions.

---

> ### Author Response · Authors · 2025-08-04
>
> We sincerely appreciate your time and thoughtful evaluation of our work. If you have any remaining comments, concerns, or suggestions, we would be grateful for the opportunity to address them in our revision.
>
> Thank you once again for your valuable contribution to the review process and your service to the research community.

---

> > ### Comment · Reviewer_qsAw · 2025-08-05
> >
> > Thanks for your response and insights into the concerns raised. However, since some of the rebuttal content was submitted beyond the valid rebuttal period, I will maintain the current score in accordance with NeurIPS rules.

---

> ### Author Response · Authors · 2025-08-05
>
> We want to thank the reviewer for the questions and feedback. We would like to clarify that all of our responses were prepared and completed within the valid rebuttal period; but due to the length limit imposed by the system, we were unable to include responses to every point in our initial message. We agree with the reviewer to strictly obey the NeurIPS rules.

---

### Official Review · Reviewer_KrZM · 2025-07-03

**Clarity:** 3
**Significance:** 3
**Originality:** 3
**Rating:** 4
**Confidence:** 4

**Summary:**

This paper introduces Adv-BMT, a framework for generating diverse and realistic safety-critical traffic scenarios by augmenting real-world driving data. The core of the framework is a novel Bidirectional Motion Transformer (BMT) that can predict agent trajectories both forwards and backwards in time. The generation process is a two-stage pipeline: it first defines a future collision state between the ego vehicle and a new adversarial agent (adversarial initialization) and then uses the BMT's reverse-prediction capability to reconstruct the adversarial trajectory leading up to that collision. This approach allows the framework to generate a high rate of successful collisions without needing any prior collision data for training.

**Questions:**

See weaknesses.

**Ethical Concerns:**

["NO or VERY MINOR ethics concerns only"]

**Final Justification:**

My concerns are mostly addressed. I encourage the authors to incoporate additional experiments in the final version.This work represents a meaningful step forward in the field of safety-critical scenario generation.

**Quality:**

3

**Strengths And Weaknesses:**

**Strengths**:
- **Novel and highly effective generation method**: The core concept of defining a collision and predicting backwards is a creative and powerful way to ensure a high rate of adversarial events. This approach effectively sidesteps the common challenge of data scarcity for safety-critical events. The method is highly effective, achieving a high adversarial collision rate and being significantly faster than baselines.
- **Value for downstream AD training**: The paper provides strong evidence that the generated scenarios are not just adversarial but also useful for improving agent performance. An RL agent trained on the dataset augmented by Adv-BMT showed superior safety, reducing collision rates by 20% compared to agents trained with data from previous methods. This demonstrates the practical value of the framework for creating more robust AD systems.

**Weaknesses:**
- **Realism of "inserted" adversaries**: The framework's primary method is to insert a new adversarial agent into an existing scene. This raises questions about the semantic realism of the scenario, as the adversary does not originate naturally from the existing traffic flow. While rule-based filters ensure kinematic plausibility, the agent might still behave in ways that are inconsistent with a typical driver's intent in that specific context. Also, I am wondering if the generated scenarios are all solvable.
- **Limited reactivity of background traffic**: In the primary reverse-prediction pipeline, the trajectories of existing background vehicles are "teacher-forced" to remain unchanged. This means the surrounding traffic does not react to the new, aggressive adversarial agent, which is unrealistic. While an optional "forward refinement" step is proposed to address this, the core method generates scenarios with limited multi-agent interactivity.
- **Limited scope of adversarial scenarios**: As acknowledged by the authors, the framework currently only considers adversarial behaviors among vehicle agents. It does not generate scenarios involving other critical road users like pedestrians or cyclists, which are essential for comprehensive safety validation of any AD system.

---

> ### Author Rebuttal · Authors · 2025-07-31
>
> We appreciate the reviewer's valuable concerns and questions.
>
> >1. The framework's primary method is to insert a new adversarial agent into an existing scene. This raises questions about the semantic realism of the scenario, as the adversary does not originate naturally from the existing traffic flow. While rule-based filters ensure kinematic plausibility, the agent might still behave in ways that are inconsistent with a typical driver's intent in that specific context.
>
> Adv-BMT is designed with multiple generation modes to support varying levels of agent interactions and realism. By default, traffic agents follow their recorded trajectories to preserve consistency with the real-world driving log, while the adversarial agent is generated to interact within this fixed context. This enables targeted scenario editing while maintaining overall scene plausibility. To address limitations in interactivity, Adv-BMT also supports a closed-loop reverse prediction mode, in which all agents are jointly predicted to generate a fully reactive traffic scenario, without teacher-forcing any agent behaviors during predictions. Additionally, an optional forward refinement step allows traffic agents to respond to the newly introduced adversary, enabling a more interactive and dynamically consistent outcome. Together, these modes allow Adv-BMT to balance realism, controllability, and diversities for different use cases.
>
>
> >2. Also, I am wondering if the generated scenarios are all solvable.
>
> We interpret "solvable" to mean that the autonomous agent is able to complete the scenario without violating safety constraints, such as collisions or failures to reach the destination. As shown in Table 3 (open-loop RL) and Table 4 (closed-loop RL), agents trained on Adv-BMT scenarios consistently achieved lower collision rates and reduced trajectory costs compared to baseline methods. Specifically, in the open-loop setting, our agent outperformed the best baseline by 5% in average cost, 14% in collision rate, and 17% in average trajectory cost. In the closed-loop setting, we also observed improvements of 5% in average cost and 1% in average trajectory cost over the best baseline. These improvements indicate that the scenarios generated by Adv-BMT are learnable and solvable by an RL agent. Furthermore, the completion rates reported in Table 3(b) and Table 4(b) show that at least 65% of the Adv-BMT scenarios are successfully completed by the agent. These results together suggest that Adv-BMT produces challenging yet solvable scenarios that are valuable for developing safer driving policies.
>
>
> >3. Limited reactivity of background traffic: In the primary reverse-prediction pipeline, the trajectories of existing background vehicles are "teacher-forced" to remain unchanged. This means the surrounding traffic does not react to the new, aggressive adversarial agent, which is unrealistic. While an optional "forward refinement" step is proposed to address this, the core method generates scenarios with limited multi-agent interactivity.
>
> Adv-BMT is designed with flexible generation modes that support varying levels of multi-agent interactions:
>
> **Adv-BMT with traffic agent teacher-forced**: In this generation mode, traffic agents are teacher-forced with their recorded trajectories. The new adversarial agent interacts with replayed traffic agents in the driving log to keep the high scene fidelity by preserving the original traffic flow.
>
> **Adv-BMT with closed-loop reverse prediction**: all agents including traffic participants and the adversarial agent are predicted jointly. It allows the entire scene to evolve coherently, enabling multi-agent interactions and producing fully synthesized scenarios with ego collisions.
>
> **Adv-BMT with forward refinement**: To introduce reactivity without sacrificing the structure of the recorded scene, this mode performs an additional forward-prediction step after generating the adversarial agent in teacher-forced mode. This allows background agents to adjust their behavior in response to the adversarial agent’s presence.
>
> These complementary modes offer a trade-off between scene realism, controllability, and interactivity, enabling users to select the appropriate setting based on their evaluation needs.
>
> **Ablation Study for Adv-BMT with baselines**:
>
> a. Realism of generated Adversarial behavior
> | Method              | Adversarial Attack Success | avg SFDE | min SFDE | avg SADE | min SADE | min Agent Collision | avg Agent Collision | min ADV-Traffic Collision | avg ADV-Traffic Collision |
> |---------------------|----------------------------|----------|----------|----------|----------|---------------------|---------------------|---------------------------|---------------------------|
> | CAT                 | 0.48                       | 1.01     | 1.01     | 0.31     | 0.31     | 0.1                 | 0.1                 | 0.3                       | 0.3                       |
> | SEAL                | 0.55                       | 1.26     | 1.26     | 0.38     | 0.38     | 0.12                | 0.12                | 0.32                      | 0.32                      |
> | STRIVE              | 0.38                       | 0.63     | 0.63     | 0.17     | 0.17     | 0.09                | 0.09                | 0.32                      | 0.32                      |
> | BMT_TF              | 1                          | 2.27     | 1.67     | 1.51     | 1.3      | 0.13                | 0.15                | 0.22                      | 0.39                      |
> | BMT_All             | 1                          | 2.36     | 1.63     | 1.56     | 1.29     | 0.13                | 0.15                | 0.19                      | 0.37                      |
> | BMT_TF_with_Filter  | 1                          | 2.28     | 1.35     | 1.53     | 1.15     | 0.13                | 0.14                | 0.12                      | 0.33                      |
> | BMT_All_with_Filter | 1                          | 1.97     | 1.23     | 1.4      | 1.09     | 0.12                | 0.14                | 0.09                      | 0.33                      |
> |                     |                            |          |          |          |          |                     |                     |                           |                           |
> |                     |                            |          |          |          |          |                     |                     |                           |                           |
> |                     |                            |          |          |          |          |                     |                     |                           |                           |
>
>
> b. Diversity of generated Adversarial behavior
> | Method              | Final Displacement Diversity | Average Displacement Diversity | Velocity JSD | Acceleration JSD | Time-To-Collide JSD |
> |---------------------|------------------------------|--------------------------------|--------------|------------------|---------------------|
> | CAT                 | 0                            | 0                              | 0.13         | 0.16             | 0.12                |
> | SEAL                | 0                            | 0                              | 0.18         | 0.25             | 0.17                |
> | STRIVE              | 0.01                         | 0                              | 0.11         | 0.22             | 0.12                |
> | BMT_TF              | 1.22                         | 0.49                           | 0.1          | 0.17             | 0.13                |
> | BMT_All             | 1.51                         | 0.63                           | 0.1          | 0.17             | 0.08                |
> | BMT_TF_with_Filter  | 2.32                         | 1                              | 0.08         | 0.12             | 0.09                |
> | BMT_All_with_Filter | 1.97                         | 0.83                           | 0.13         | 0.14             | 0.08                |
>
> (Note: here the min/avg refers to the mode selection method for each scene)
>
> **Analysis**: Comparing adversarial agents generated under different modes (BMT_TF for teacher-forced traffic agents, and BMT_All for fully interactive generation), we observe a trade-off between realism and diversity. Specifically, BMT_TF generated ADV trajectories have slightly better realism-related metrics (lower ADE/FDE), while BMT_All exhibits better behavioral diversity and interaction, as indicated by greater displacement diversity metrics and lower ADV-traffic collision rates.
>
> We appreciate the reviewer’s questions and concerns, and we will emphasize in revision that Adv-BMT is designed to flexibly support multiple interaction modes, enabling a controllable balance between realism and diversity based on application needs.
>
>
> >4. Limited scope of adversarial scenarios: As acknowledged by the authors, the framework currently only considers adversarial behaviors among vehicle agents. It does not generate scenarios involving other critical road users like pedestrians or cyclists, which are essential for comprehensive safety validation of any AD system.
>
> Thank you for raising this important point. We would like to clarify that Adv-BMT does in fact support the generation of adversarial pedestrian and bicycle agents, enabled by the fact that our prediction model has been trained on scenarios involving these agent types. Specifically, adversarial pedestrians can be initialized in scenes where the ego vehicle’s trajectory is close to existing crosswalks on the map, limiting their applicability to a subset of maps. In contrast, adversarial bicycle agents can be initialized more flexibly across a wider range of scenes. We appreciate the reviewer’s feedback and will revise the manuscript accordingly by updating the limitation section and adding visualizations to better reflect this capability.

---

> > ### Comment · Reviewer_KrZM · 2025-08-04
> >
> > Thanks for the rebuttal! I have read the response and other reviews carefully. Most of my concerns are well addressed. The additional experiments demonstrate the flexibility of the proposed framework. I would like to keep my score and recommend acceptance.

---

> ### Author Response · Authors · 2025-08-04
>
> We would like to thank the reviewer. As the authors, we greatly appreciate the time and effort you dedicated to evaluating our work.
>
> As we are committed to continuously improving the quality and clarity of our work, we would sincerely welcome any further feedback or clarification you may be willing to share. For any remaining concerns or suggestions that were not addressed during the rebuttal phase, we would be grateful for the opportunity to consider them and incorporate your insights into our revision.
>
> Thank you again for your contribution to the review process and for your service to the community.

---

### Note · Authors · 2025-08-14

Through the rebuttal and discussion process, we have made every effort to address the reviewers’ concerns.

**Reviewer KrZM**

For semantic realism, we clarified that Adv-BMT offers multiple generation modes (default teacher-forced traffic, closed-loop reverse prediction, and forward refinement) to balance realism, controllability, and diversity.

For limited reactivity, we explained how the closed-loop and forward-refinement modes enable coherent multi-agent interactions beyond fixed background traffic, supported by an ablation study showing the trade-offs between realism and diversity.

Regarding scope limitation, we corrected that Adv-BMT supports adversarial pedestrian and bicycle agents. We will provide qualitative visualizations in our revision to support this claim.

**Reviewer qsAw**

As clarifications, we confirmed the “Bidirection” setting and showed that Bidirection predictions improve over both realism and diversity scores compared to reverse and forward predictions.

For ablation studies, we clarified the effect of our filtering improves realism metrics without compromising diversity.

We incorporated the revision suggestions to improve clarity and ensure fair baseline comparisons, and we reaffirmed our commitment to releasing the code/checkpoint in September.

The reviewer only responded to the timing of our rebuttal threads, which constrained from the strict character limit for the rebuttal window.

**Reviewer v8s5**

We explained the RL improvement margins, showing consistent gains in collision rate and trajectory cost over baselines.

We analyzed why reverse prediction yields more unsafe behaviors: most collisions in reverse predictions with real initializations stem from crowded parking lot scenes where nearby static agents are flagged as collisions, explaining comparable ADE/FDE despite higher collision scores. The performance gap reflects our training procedure: pre-trained on forward prediction before bidirectional tasks.

We discussed the advantage of using acceleration/yaw rate tokenization over k-disk from previous work and incompatibility for bidirectional predictions.

We haven't heard from the reviewer from our follow-up response and message.

**Summary**

By introducing the first bidirectional motion transformer with diverse collision initializations, Adv-BMT achieves both high realism and diversity in generated safety-critical traffic behaviors, delivering improvements in autonomous driving safety and robustness.

---

### Decision · Program_Chairs · 2025-09-17

**Decision:**

Accept (poster)

**Comment:**

The paper introduces Adv-BMT, a framework for generating safety-critical traffic scenarios using a bidirectional motion transformer. It reconstructs trajectories in reverse from adversarial collision states, enabling diverse and realistic scenario generation without collision data pretraining. Results show improved diversity, realism, and 20% lower collision rates for RL agents trained with augmented data.

Strengths: creative use of reverse prediction; strong empirical results across realism, diversity, and downstream safety; flexible modes (teacher-forced, closed-loop, forward refinement) balancing realism and diversity; thorough ablations; code release planned.

Weaknesses: realism of adversarial agents questioned; limited background traffic reactivity; scope mainly focused on vehicles, with limited validation on pedestrians/cyclists; some fairness concerns due to rule-based filtering; clarity of writing and comparisons could be improved.

In the rebuttal, authors clarified bidirectional prediction, detailed teacher-forcing strategies, added ablations on filtering and initialization, and emphasized support for pedestrian/cyclist adversaries. Reviewers found concerns on realism and fairness partly resolved. While novelty and clarity were noted as moderate, the demonstrated utility and comprehensive evaluation led reviewers to lean positive.

Despite limitations, the work addresses a key challenge in AD safety testing, proposes a novel and effective solution, and demonstrates strong practical impact through improved robustness of trained agents.